# FAM21 is critical for TLR2/CLEC4E-mediated dendritic cell function against *Candida albicans*

Rakesh Kulkarni[1,2,*], Siti Khadijah Kasani[1,2,*], Ching-Yen Tsai[2], Shu-Yun Tung[2], Kun-Hai Yeh[2], Chen-Hsin Albert Yu[2], Wen Chang[2]

**FAM21 (family with sequence similarity 21) is a component of the Wiskott–Aldrich syndrome protein and SCAR homologue (WASH) protein complex that mediates actin polymerization at endosomal membranes to facilitate sorting of cargo-containing vesicles out of endosomes. To study the function of FAM21 in vivo, we generated conditional knockout (cKO) mice in the C57BL/6 background in which FAM21 was specifically knocked out of CD11c-positive dendritic cells. BMDCs from those mice displayed enlarged early endosomes, and altered cell migration and morphology relative to WT cells. FAM21-cKO cells were less competent in phagocytosis and protein antigen presentation in vitro, though peptide antigen presentation was not affected. More importantly, we identified the TLR2/CLEC4E signaling pathway as being down-regulated in FAM21-cKO BMDCs when challenged with its specific ligand *Candida albicans*. Moreover, FAM21-cKO mice were more susceptible to *C. albicans* infection than WT mice. Reconstitution of WT BMDCs in FAM21-cKO mice rescued them from lethal *C. albicans* infection. Thus, our study highlights the importance of FAM21 in a host immune response against a significant pathogen.**

## Introduction

Endocytosis is defined as the uptake of components on the plasma membrane and in extracellular fluids into the cytoplasm (Kaksonen & Roux, 2018). Several types of endocytosis have been identified, including clathrin-mediated endocytosis, caveolae, macropinocytosis, phagocytosis, and clathrin-independent pathways, which differ depending on the size and type of cargo that is endocytosed (Doherty & McMahon, 2009). Endocytosis mediates important biological processes such as nutrient uptake, cell signaling, antigen presentation, cell adhesion, migration, and mitosis (Doherty & McMahon, 2009). Pathogens such as bacteria and viruses exploit the cellular endocytic machinery to enter host cells (Bonazzi & Cossart, 2006; Brass et al, 2008; Cossart & Helenius, 2014). Several studies have highlighted the role of the actin cytoskeleton in regulating endocytosis such as via recruitment of adaptor proteins, concentration of cargo, stabilization of endosomal domains, and formation of phagocytic cups (Galletta & Cooper, 2009; Kumari et al, 2010; Simonetti & Cullen, 2019). Depending on their function, cargoes present in endosomes are directed toward their respective destinations (Mooren et al, 2012; Deng et al, 2015). Mechanisms of endocytosis and cellular trafficking are orchestrated by multi-protein assemblies that include retromer and retriever, sorting nexins, and the actin-related protein 2/3 (ARP2/3)–activating WASH complex (Seaman, 2012; Simonetti & Cullen, 2019; Fokin & Gautreau, 2021). Our previous studies on HeLa cells have shown that vaccinia mature virus (VACV) is endocytosed into host cells and that a novel protein, FAM21—also known as vaccinia virus penetration factor—is required for virus penetration (Hsiao et al, 2015). KD of FAM21 expression blocked intracellular transport of dextran and VACV penetration into cells but did not inhibit their binding to cells, suggesting that FAM21 mediates the VACV endocytic process in HeLa cells (Huang et al, 2008).

FAM21 is a large 1334-aa protein, containing a globular "head" domain (~200 aa) at its N-terminus and a long (~1,100 aa) C-terminal "tail" that contains a binding site for the actin-capping protein CapZ (Gomez & Billadeau, 2009; Jia et al, 2012). FAM21 is a component of the WASH protein complex that also comprises WASH1, strumpellin, strumpellin and WASH-interacting protein (SWIP), and coiled-coil domain–containing protein 53 (CCDC53) (Harbour et al, 2012; Jia et al, 2012). The WASH protein complex interacts with endosomes through the binding of its constituent FAM21 to the vacuolar protein sorting retromer complex (consisting of VPS26, VPS29, and VPS35) anchored on endosomal membranes (Harbour et al, 2012). The WASH protein complex then recruits ARP2/3 to mediate actin polymerization at endosomal membranes so that cargo-containing vesicles can be budded out of endosomes (Gomez & Billadeau, 2009). FAM21 is also required for trafficking of receptor proteins essential for nutrient uptake, such as the glucose transporter GLUT1 (Lee et al, 2016) and the copper transporter copper metabolism MURR1 domain–containing 1 (COMMD1) (Phillips-Krawczak et al, 2015). When impaired, disruption of these receptor proteins results in deficiencies of essential cell nutrients.

[1]Molecular and Cell Biology, Taiwan International Graduate Program, Academia Sinica and Graduate Institute of Life Science, National Defense Medical Center, Taipei, Taiwan [2]Institute of Molecular Biology, Academia Sinica, Taipei, Taiwan

Correspondence: mbwen@ccvax.sinica.edu.tw
*Rakesh Kulkarni and Siti Khadijah Kasani contributed equally to this work

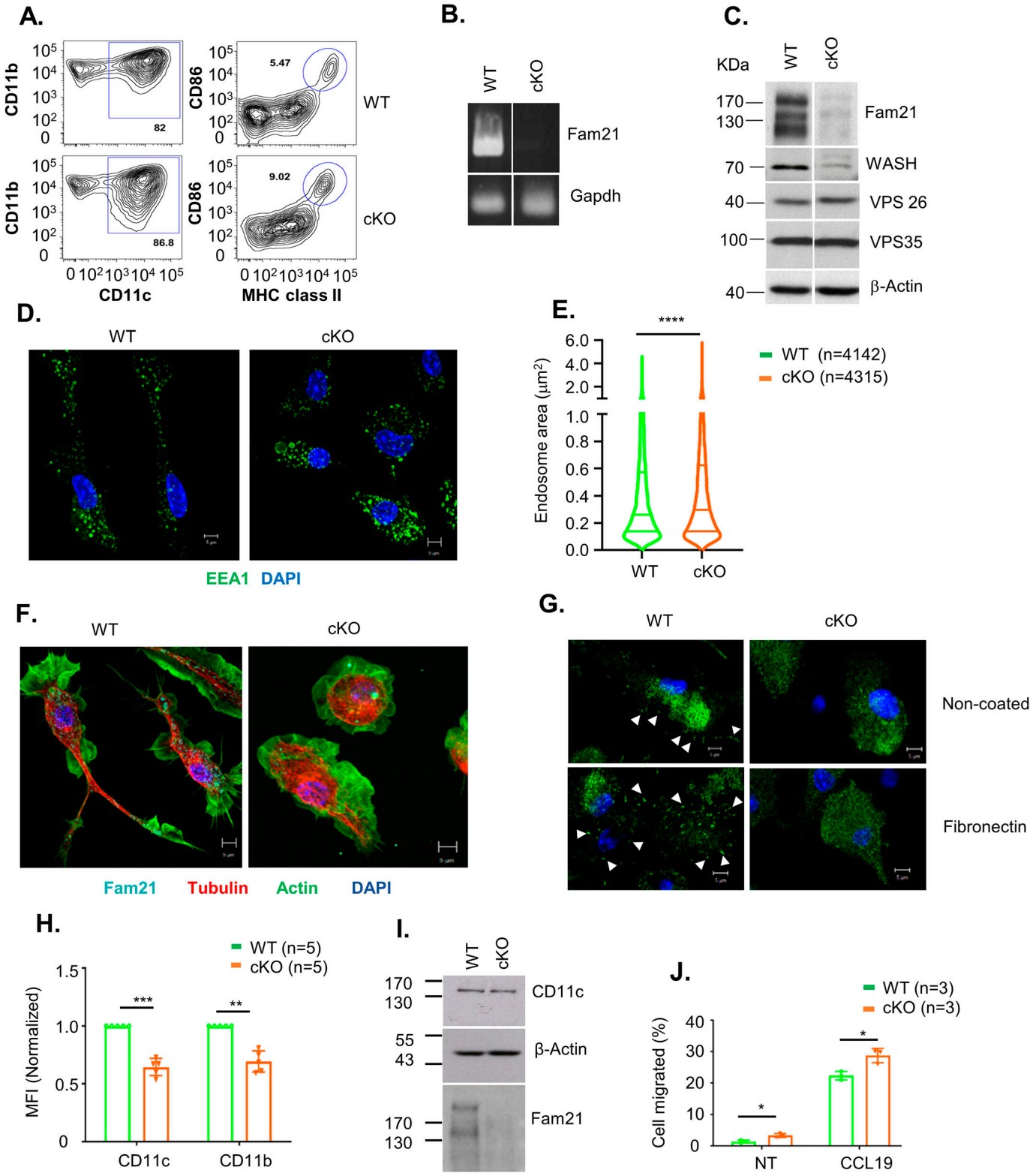

**Figure 1. FAM21-cKO BMDCs present enlarged early endosomes and exhibit reduced cell spreading ability.**
**(A)** Flow cytometry of BMDCs isolated at day 8 post-isolation from WT and FAM21-cKO mice using anti-CD11b, anti-CD11c, anti-CD86, and anti-MHC class II antibodies.
**(B)** RT–PCR of FAM21 transcript in WT and cKO BMDCs. GAPDH was used as a control. **(C)** Immunoblot of FAM21 in BMDCs cultured from WT and FAM21-cKO mice. WASH and the retromer complex proteins VPS26 and VPS35 are also shown. $\beta$-Actin was used as a control. **(D)** Immunofluorescence staining of early endosomes (green) and nuclei (DAPI, blue) in WT and cKO BMDCs. **(E)** Quantification of endosome size distributions in WT and cKO BMDCs using violin plots. n = total number of endosomes measured for each genotype. ****$P < 0.0001$. **(F)** Confocal images of WT and cKO BMDCs stained with anti-FAM21 (cyan), anti-tubulin (red), phalloidin (green), and DAPI (blue).

Although most FAM21 functions in endosomal sorting and trafficking are dependent on the WASH protein complex (Gomez & Billadeau, 2009), previous studies have shown that FAM21 also exerts WASH-independent functions (Deng et al, 2015; Lee et al, 2016). Studies have shown that FAM21, but not the WASH complex, could enhance NF-κB transcriptional activation (Deng et al, 2015; Lee et al, 2016). Another study by Lee et al (2016) showed that FAM21 regulates PI4KB levels at Golgi independently of the WASH complex (Lee et al, 2016). In mammalian cells, but not in *Dictyostelium* (Park et al, 2013), formation of the intact WASH complex dictates the stability of each of its subunits (Derivery et al, 2009; Jia et al, 2012), and it has been shown that an E3 RING ubiquitin ligase, MAGE-L2-TRIM27, targets the WASH protein for degradation (Hao et al, 2013). K63 ubiquitination of WASH at residue K220 is facilitated by MAGE-L2-TRIM27, which results in F-actin nucleation and retromer-dependent transport (Hao et al, 2013).

The trafficking defects derived from mutations or loss of functional subunits in the WASH complex result in a wide range of human diseases. The WASH complex has been demonstrated as important for cell invasion by *Salmonella* (Unsworth et al, 2004; Hanisch et al, 2010). Mutations in strumpellin cause autosomal dominant hereditary spastic paraplegia, a neurodegenerative disorder of upper neurons (Valdmanis et al, 2007). Homozygous P1019R mutation in SWIP causes autosomal recessive mental retardation 43 (MRT-43) (Ropers et al, 2011). A point mutation (D620N) within the VPS35 subunit of the vacuolar protein sorting retromer complex, with which the WASH complex directly interacts, is associated with Parkinson's disease and is also responsible for disrupting cargo sorting (Follett et al, 2014). Previous studies have also shown that KO of WASH, strumpellin, and the retromer protein subunit VPS26a all individually result in embryonic lethality (Muhammad et al, 2008; Piotrowski et al, 2013; Tyrrell et al, 2016), though VPS26b-KO mice survived despite displaying sortilin trafficking defects (Kim et al, 2010). Genetic disruption arising from SWIP[P1019R] mutation resulted in significant and progressive motor deficits in mice that were similar to movement deficits observed in humans (Courtland et al, 2021). Despite being a component of the WASH protein complex, whether FAM21 mutations are associated with any known human diseases remains unknown, so our investigation of the function of FAM21 in vivo is critical to understanding whether FAM21 plays a role in disease prevention.

## Results

### FAM21 cKO BMDCs exhibit enlarged early endosomes, and reduced cell spread and migration

To investigate the role of FAM21 in vivo, we generated a constitutive FAM21 KO mouse line (Fig S1A), but no $FAM21^{-/-}$ littermates were obtained. FAM21 deficiency caused embryonic lethality as early as E7.5 (Fig S1B), evidencing an essential role of the FAM21 protein in embryonic development. Northern blot analyses revealed that FAM21 is widely expressed in multiple organs, including brain, intestine, kidney, liver, lungs, lymph nodes, and thymus (Fig S1C). FAM21 is also expressed in BMDCs (Fig S1C), and DC plays an essential role in innate immunity such as pathogen sensing, cytokine secretion, and antigen presentation to activate adaptive immune responses (Eisenbarth, 2019; Cabeza-Cabrerizo et al, 2021). Furthermore, depletion of immune cells or KO of genes in the immune system often does not cause lethality in mice, allowing us to obtain adult KO mice for phenotypic studies. We also rationalized that many DC functions rely on cytoskeleton rearrangement and cell polarization, which may be involved in the cargo trafficking activity modulated by Fam21.

We generated $FAM21^{+/-}$ germline deletion mice in which one allele of the $FAM21$ gene had been knocked out in all somatic cells. Crossing the $FAM21^{+/-}$ mice with $FAM21$ $^{f/f\ CD11c-Cre}$ mice generated cKO BMDCs hosting only residual Fam21 protein, representing a suitable null mutant for in vitro studies. However, these germline $FAM21^{+/-}$ mice are unsuited to study Fam21 function in vivo because they generate offspring containing somatic cells with only one functional allele of the $FAM21$ gene. Therefore, for our purposes, we bred $FAM21^{f/f}$ mice with CD11c-Cre mice to specifically knock out the $FAM21$ gene from CD11c-positive dendritic cells, allowing us to address the biological roles of Fam21 in vivo (Fig S1F). All the mice we generated according to these two KO strategies developed normally (data not shown).

Bone marrow cells extracted from femur and tibia of WT and the cKO mice described above were cultured with GM-CSF to induce dendritic cell differentiation in vitro. At day 8, the cells were analyzed using anti-CD11b and anti-CD11c antibodies, which revealed similar levels (~85%) of CD11c$^+$CD11b$^+$ BMDCs in both WT and cKO mice. Less than 10% of the BMDC cultures expressed high levels of CD86 and MHC class II activation markers, indicating that the majority were immature BMDCs (Fig 1A). We have quantified the CD86$^{high}$MHCII$^{high}$, CD86$^{med}$MHCII$^{med}$, and CD86$^{low}$MHC$^{Low}$ populations (Fig S2), and the data show only CD86$^{high}$MHCII$^{high}$ population was up-regulated in cKO BMDC relative to WT cells. The up-regulation of CD86 and MHC-II in cKO BMDC could be due to the defect in their internalization. Both RT–PCR (Fig 1B) and immunoblot analysis (Fig 1C) confirmed that FAM21 expression was significantly reduced in FAM21-cKO BMDCs. Although FAM21 protein displayed three forms of different molecular weight, perhaps derived from alternative splicing (Fig S3), all three forms were significantly reduced in the cKO BMDCs (Fig 1C).

FAM21 is known to attach to endosomes by binding to retromers, and it recruits WASH to facilitate retrograde transport of vesicles through scission of cargo-enriched subdomains of early endosomes (Gomez & Billadeau, 2009; Jia et al, 2010; Harbour et al, 2012). We stained WT and FAM21-cKO BMDCs with anti-EEA1 (Fig 1D),

---

**(G)** Confocal images of WT and cKO BMDCs cultured on non-coated or fibronectin-coated (10 μg/ml) glass slides and stained with anti-paxillin (green) and DAPI (blue). White arrows indicate focal adhesion points. **(H)** Quantification (as mean fluorescence intensity, MFI) of CD11c and CD11b integrin receptors on the surface of WT and cKO BMDCs (n = 5 per genotype). Data are represented as the mean ± SD. **P < 0.01 and ***P < 0.001. **(I)** Immunoblot of CD11c, FAM21, and actin in total cell lysates of WT and cKO BMDCs. **(J)** Transwell migration assay (12-well) of WT and cKO BMDCs. Cells were seeded in the upper compartment of a transwell plate, and the chemokine CCL19 was added in the lower compartment (300 ng/ml) or non-treated (NT) and incubated for 3 h at 37°C. Cells that migrated to the lower compartment were collected and counted (n = 3 per genotype). The scale bar represents 5 μm. Data are represented as the mean ± SD. *P < 0.05.
Source data are available for this figure.

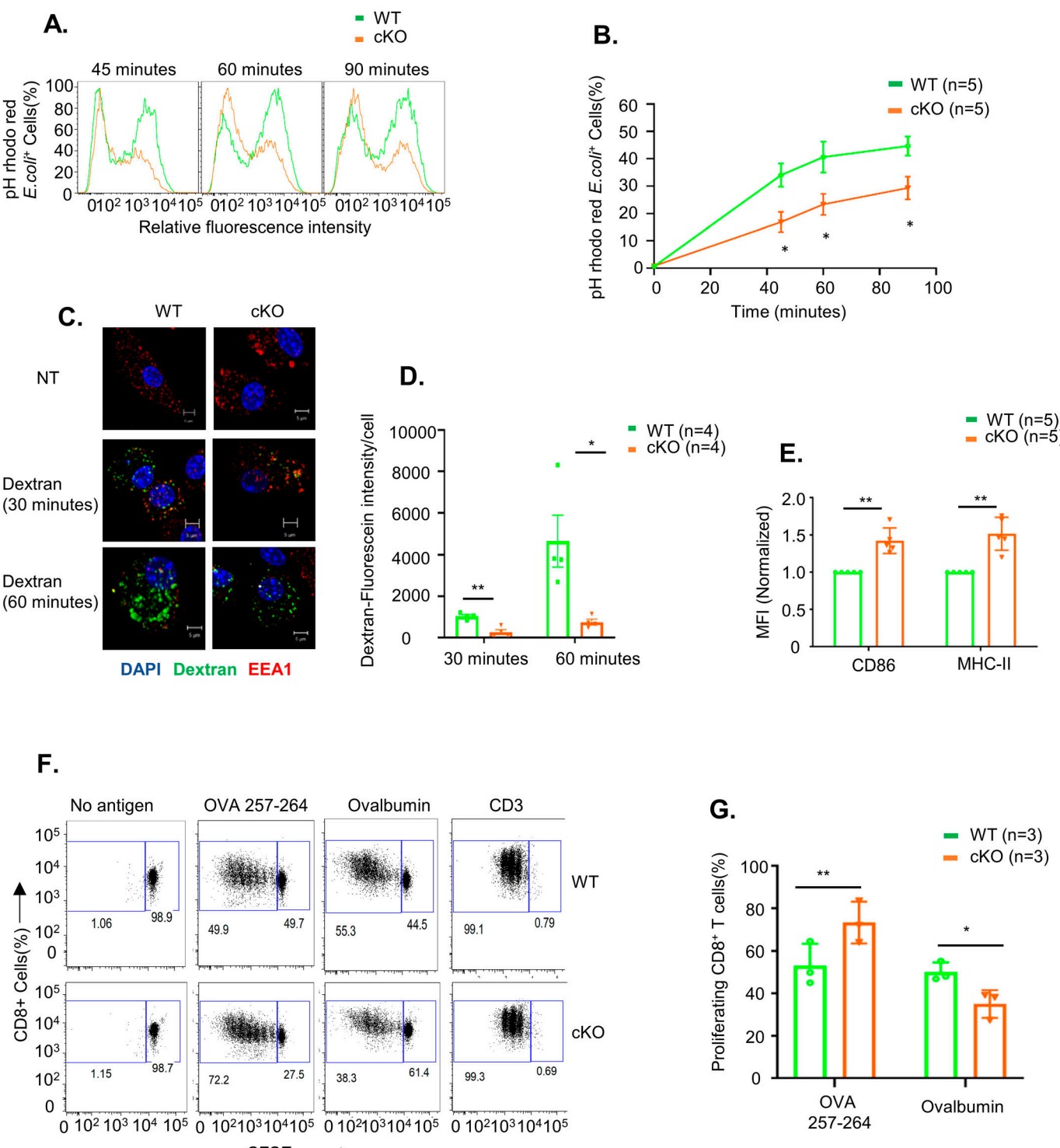

**Figure 2. FAM21 is important for phagocytosis and the protein antigen presentation function of BMDCs.**
**(A)** Fluorescence intensity of pHrodo Red *E. coli* taken up by WT and cKO BMDCs at 45, 60, and 90 min after adding bacteria to cells. **(B)** Percentage of BMDCs containing phagocytosed pHrodo Red *E. coli* from 0 to 90 min (n = 5 per genotype). Data are represented as the mean ± SEM. *$P$ < 0.05. **(C)** Imaging analysis of dextran macropinocytosis in WT and cKO BMDCs. Cells were incubated with fluorescent dextran (green) (1 μg/ml) for 30 and 60 min at 37°C, fixed, and then stained with anti-EEA1 (red) and DAPI (blue). NT; non-treatment control. **(D)** Measurement of dextran–fluorescein uptake by WT and cKO BMDC cells quantified by ImageJ software at 30 and 60 min, (n = 4 per genotype at each time point, ~40 cells quantified for each genotype and each time point). **(E)** Quantification of MFI of co-stimulatory molecules CD86 and MHC class II on the surface of WT and cKO BMDCs. Data are represented as the mean ± SD. **$P$ < 0.01 (n = 5 per genotype). **(F)** Antigen presentation assay on WT and cKO BMDCs. CFSE-labeled CD8⁺ T cells isolated from OT-1 transgenic mice were co-cultured with WT and cKO BMDCs that had undergone prior pulsing with either ovalbumin protein or OVA 257–264 peptide. After 48 h of co-culture, CFSE dye dilution on CD8⁺ OT-1 cells was analyzed as an index of T-cell proliferation. CD3 served as a positive

measured ~4,000 endosomes in each cell population, and concluded that the early endosomes of cKO BMDCs were significantly enlarged relative to WT cells (Fig 1E). We speculate that this outcome could be derived from an endosome fission defect in the cKO cells, similar to our previous observation for *FAM21*-KO HeLa cells (Hsiao et al, 2015).

Upon seeding WT and cKO BMDCs onto glass slides for imaging analyses, we noticed that most of the cKO cells remained unattached even after 16 h, whereas WT BMDCs were well attached. Hence, we tested the adhesion ability of both WT and cKO BMDC cells using crystal violet staining and observed WT BMDC attached quickly on both uncoated and fibronectin-coated wells compared with cKO BMDC cells, implying an adhesion defect of cKO cells (Fig S4). Accordingly, we stained both cell types for cytoskeletal actin and α-tubulin, which revealed clear morphological differences between the WT and cKO cells (Fig 1F), with WT BMDCs being elongated and displaying cell polarity, whereas cKO cells were rounded. Next, we seeded both cell types onto slides coated with fibronectin and stained for focal adhesions using anti-paxillin antibody, which demonstrated that cKO cells hosted fewer focal adhesion points than WT (Fig 1G). We also noticed that surface expression of two integrin markers, CD11b and CD11c, was reduced in cKO cells relative to WT cells (Fig 1H), even though respective protein amounts in WT and cKO cell lysates were comparable (Fig 1I), implying an integrin receptor trafficking defect. We performed the internalization assay twice for the CD11c and CD11b receptors, respectively, by means of flow cytometry and observed that both receptors were internalized faster in the cKO BMDCs relative to the WT cells (Fig S5). Despite this difference in internalization kinetics, we cannot rule out a contribution of other recycling defects to our observations for the FAM21-cKO cells. Moreover, a cell migration assay revealed that cKO BMDCs moved faster than WT cells in the presence or absence of the chemokine CCL19 (Fig 1J). Taken together, these results indicate that FAM21-cKO cells exhibit altered cell morphology, diminished cell adhesion ability, and enhanced cell migration.

## FAM21 is important for the phagocytic activity of BMDCs

The immune-related functions of dendritic cells rely on their ability to take up foreign antigens, internally process them, and present them to other immune cells by displaying them on their surface. We tested whether FAM21-cKO affected the phagocytic function of BMDCs. WT and FAM21-cKO BMDCs were incubated with pHrodo Red–labeled *Escherichia coli* particles, and then, we analyzed (Fig 2A) and quantified (Fig 2B) the amounts of bacteria engulfed into endosomes by FACS at 45, 60, and 90 min after incubation, which showed that cKO cells displayed a reduced ability to phagocytose bacteria. Macropinocytic uptake of fluorescent dextran involves cellular engulfment of solutes from medium, and we also adopted this confocal microscopy approach at 30 and 60 min to show that dextran was internalized less efficiently in FAM21-cKO BMDCs compared with WT BMDCs (Fig 2C) quantified in (Fig 2D). Thus, we

conclude that FAM21 is an important regulator of the endocytic activity of BMDCs.

We determined that the co-stimulatory molecules CD86 and MHC class II were up-regulated in cKO BMDCs relative to WT BMDCs (Fig 2E). Hence, we investigated whether FAM21 cKO affects the antigen presentation function of BMDCs. To do so, we used two antigen types: ovalbumin that needs to be internalized, processed, and loaded onto the co-stimulatory molecules of dendritic cells for presentation to T cells; and the small peptide OVA 257–264 that can be exogenously loaded onto cell surface co-stimulatory molecules for direct presentation to T cells. First, we pulsed BMDCs with ovalbumin or OVA 257–264, and then co-cultured them with CD8[+] T cells isolated from OT-1 transgenic mice. CD8[+] T-cell proliferation was then monitored by FACS (Fig 2F) (Quah et al, 2007). We found that WT BMDCs co-cultured with ovalbumin or OVA 257–264 stimulated robust CD8[+] T-cell proliferation. In contrast, whereas FAM21-cKO BMDCs pulsed with OVA 257–264 stimulated CD8[+] T-cell proliferation better than WT BMDCs, they were less effective when pulsed with ovalbumin suggesting a reduced antigen uptake (Fig 2F) quantified in Fig 2G. We did not measure surface expression levels of MHC-I in WT or KO BMDC cells, but it has previously been reported that increased expression of MHC-I molecules in DCs results in enhanced peptide antigen presentation (Ackerman & Cresswell, 2003). Accordingly, it is possible that the enhanced ability of our cKO BMDCs to present peptide antigen is due to increased levels of MHC-I molecules on cell surfaces.

## Microarray analysis reveals down-regulation of the TLR2/CLEC4E signaling pathway in FAM21-cKO BMDCs

A previous study of pancreatic cancer cells revealed that FAM21 knockdown affected NF-κB–mediated gene expression (Deng et al, 2015) in a WASH-independent manner. Hence, we investigated differential gene expression in WT and FAM21-cKO BMDCs by performing an unbiased microarray analysis of RNA samples. Microarray data were analyzed in GeneSpring (version 12.6.1), which identified 98 significantly up-regulated ($\log_2$ fold change >1.5) and 148 down-regulated (fold change <1.5) genes in FAM21-cKO BMDCs (Fig 3A and B). Next, we performed pathway enrichment analysis and identified up-regulated (Fig S6) and down-regulated (Fig 3C) pathways and, notably, that cytokine–cytokine receptor interaction was the most down-regulated pathway in the cKO BMDCs (Fig 3C). By means of STRING pathway analysis (https://string-db.org/), we discovered that several cellular transcripts in the TLR2/CLEC4E signaling pathway were down-regulated in cKO BMDCs relative to WT (Fig 3D), such as those encoding CLEC4E, TLR2, CXCL2, IL-1β, and TNF-α, as validated by qRT–PCR (Fig 3E).

## FAM21 in BMDCs is important for TLR2/CLEC4E signaling pathway activation upon *C. albicans* infection

Next, we monitored cell surface expression levels of CLEC4E by flow cytometry (Fig 4A), with quantification demonstrating reduced

control. **(G)** Quantification of CFSE-labeled CD8[+] T-cell proliferation, as described in (F) (n = 3 per genotype). The scale bar represents 5 $\mu$m. Data are represented as the mean ± SD. *P < 0.05 and **P < 0.01.
Source data are available for this figure.

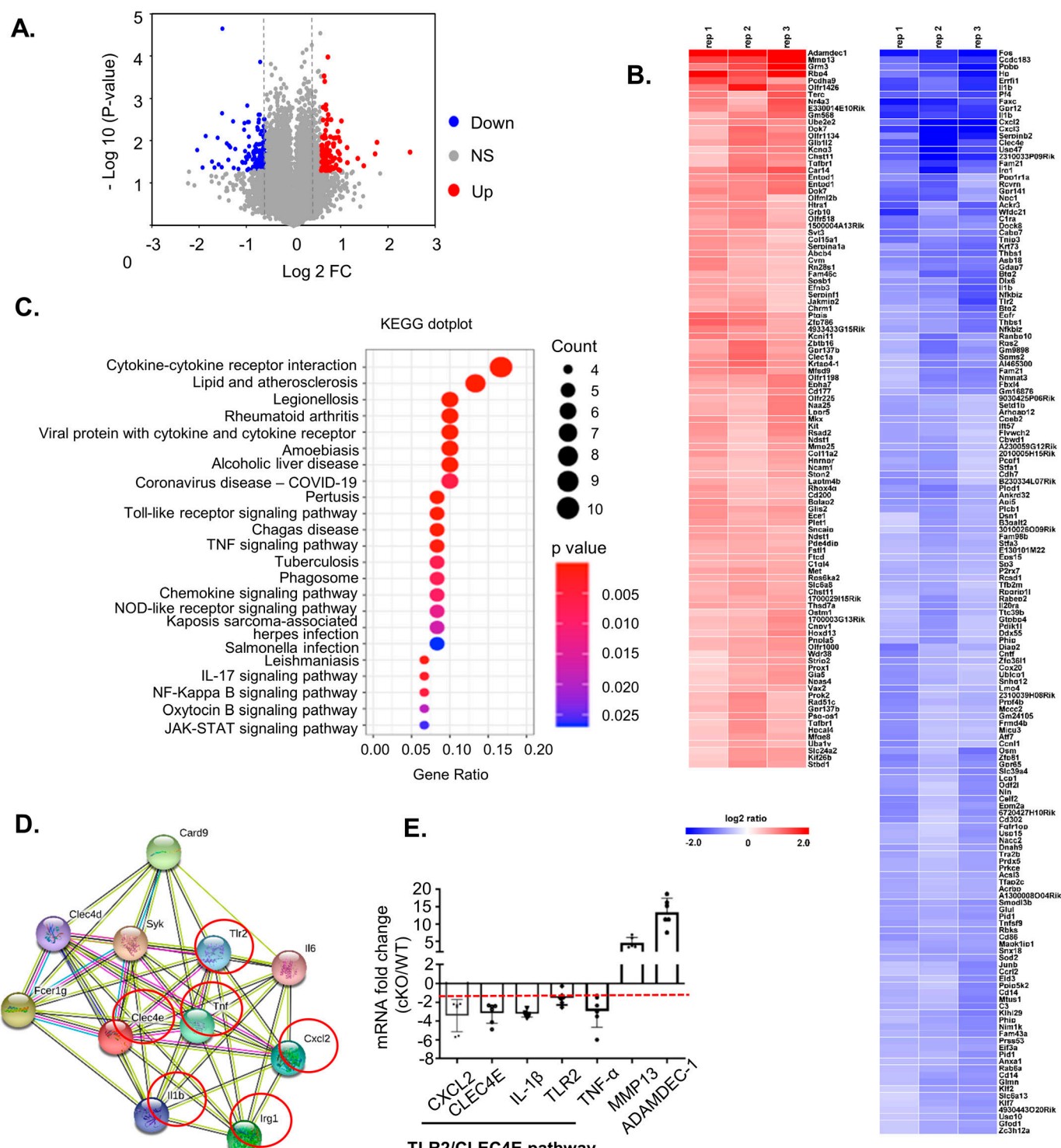

**Figure 3. Microarray analysis reveals TLR2/CLEC4E pathway down-regulation in cKO BMDCs.**
**(A)** Volcano plot depicting numbers of genes either 1.5-fold up-regulated (red) or down-regulated (blue) in cKO BMDCs relative to WT cells. All other genes are colored gray. **(B)** Heatmap of genes whose transcripts are either 1.5-fold up-regulated (red) or down-regulated (blue) in cKO BMDCs relative to WT BMDCs (n = 3 per genotype). **(C)** Gene enrichment analysis of the down-regulated genes by KEGG dot plot. **(D)** STRING (version 11.0) pathway analysis of CLEC4E-binding molecules. Red circles represent proteins that are down-regulated >1.5-fold in cKO BMDCs relative to WT cells. **(E)** qRT–PCR of RNA samples from WT and cKO BMDCs (n = 6 per genotype). The red dashed line represents a twofold reduction level. Data are represented as the mean ± SD. *$P < 0.05$.
Source data are available for this figure.

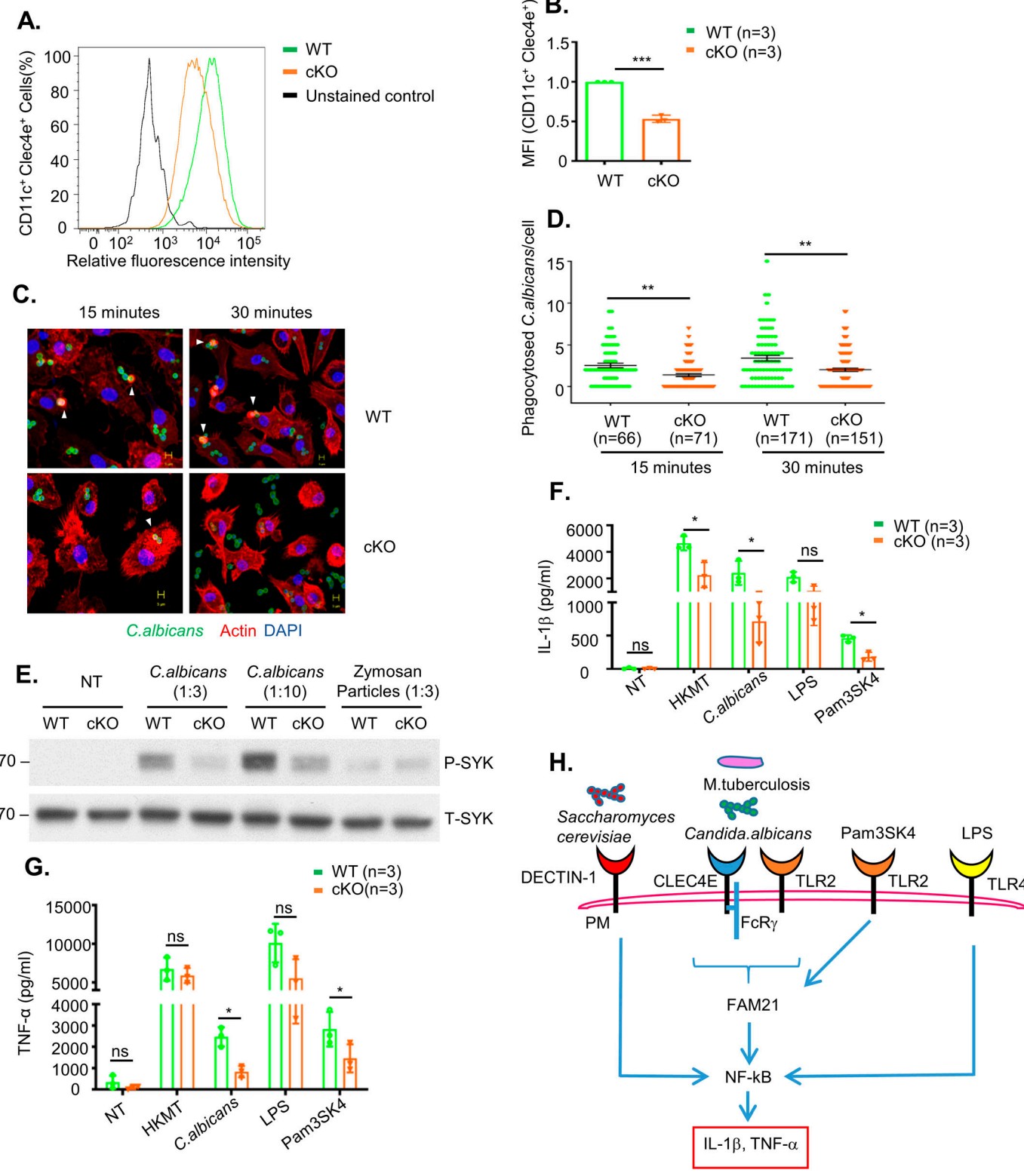

**Figure 4. TLR2/CLEC4E signaling pathway is down-regulated in cKO BMDCs upon stimulation with specific ligands.**
**(A)** Surface expression of CLEC4E in CD11c[+] BMDCs from WT and cKO mice, as determined by flow cytometry. **(A, B)** Quantification of CLEC4E MFI in WT and cKO BMDCs from (A) (n = 3 per genotype). **(C)** Representative images of phagocytosed *C. albicans* in WT and cKO BMDCs. CFSE-labeled *C. albicans* (green) was co-cultured with WT and FAM21-cKO BMDCs for 15 or 30 min, fixed with 4% paraformaldehyde, and then stained for phalloidin (red) and DAPI (blue). Arrowheads point to phagocytosed *C. albicans*. **(C, D)** Quantification of phagocytosed *C. albicans* per cell for WT and cKO BMDCs from (C) (n, numbers of cells analyzed for each genotype for each time point). Data are represented as the mean ± SEM. **P < 0.01. **(E)** Immunoblot of phosphorylated SYK (P-SYK) and total SYK (T-SYK) in WT and cKO BMDCs at 30 min after infection with

levels of CLEC4E in FAM21-cKO BMDCs (Fig 4B). CLEC4E is a cell surface receptor expressed predominantly in myeloid cells such as monocytes, macrophages, and neutrophils (Matsumoto et al, 1999; Ostrop et al, 2015; Drouin et al, 2020). Together with TLR2, it is an important component of the pathogen-sensing cellular mechanism that detects infectious agents such as *Candida albicans* and Mycobacteria, and other threats such as dead cell–derived proteins like SAP-130 (Bugarcic et al, 2008; Yamasaki et al, 2008). We infected WT and FAM21-cKO BMDCs with *C. albicans* and then measured phagocytosis of this pathogen at 15 and 30 min. Significantly, the cKO BMDCs phagocytosed much fewer *C. albicans* than WT BMDCs (Fig 4C and D). Moreover, phosphorylation of the *C. albicans*–induced downstream kinase SYK was attenuated in the cKO BMDCs relative to WT. We also stimulated the BMDCs with the ligand zymosan, which is detected by dectin-1 and activates SYK phosphorylation. However, levels of SYK phosphorylation upon zymosan treatment were similar between WT and FAM21 cKO BMDCs, suggesting CLEC4E-specific SYK down-regulation in our cKO cells (Fig 4E). In addition, we tested a panel of innate immune stimuli—including heat-killed *Mycobacterium tuberculosis* and *C. albicans* as CLEC4E/TLR2-specific ligands, LPS as a TLR4-specific ligand, and Pam3SK as a TLR2-specific ligand—and then measured the expression of downstream cytokines (Wells et al, 2008; Ostrop et al, 2015; Clement et al, 2016). Our data in general support that the CLEC4E/TLR2 signaling pathway is considerably impaired in the cKO cells, whereas the LPS-induced TLR4 signaling pathway remains intact (Fig 4F and G). Together, these findings show that the FAM21 protein is involved in the innate immune TLR2/CLEC4E signaling pathway of murine BMDCs, as summarized in Fig 4H.

### FAM21-cKO mice exhibit increased susceptibility to *C. albicans* in vivo

Next, we challenged WT and FAM21-cKO mice i.p. with *C. albicans* and monitored body weight changes for 7–8 d. The cKO mice clearly lost more weight than WT mice between days five and seven (Fig 5A), and 70% of the cKO mice had died by day 7 compared with only 20% of WT mice (Fig 5B). *C. albicans* titers in the kidneys of cKO mice were also higher than those of WT mice (Fig 5C). Moreover, H&E staining of kidneys at day 7 also revealed more *C. albicans* colonies in the cKO mice than in WT mice (Fig 5D), demonstrating that FAM21 functioning in CD11c[+] dendritic cells is critical for *C. albicans* clearance in vivo.

### FAM21-cKO mice exhibit reduced innate and adaptive immune cell infiltration at infection sites

To understand whether immune cell homeostasis is affected in FAM21-cKO mice upon *C. albicans* infection, we harvested immune cell populations from infection sites by means of i.p. washes at days 1 (Fig S7) and 3 post-infection (p.i.) for flow cytometric analyses. At day 1 p.i., we detected reduced recruitment of dendritic cells (Fig 6A), macrophages (Fig 6B), and neutrophils (Fig 6C) in the FAM21-cKO mice relative to WT, but there were no major changes in their B- and T-cell populations (Fig 6D–I). However, at 3 days post-infection (d.p.i), recruitment of B and T cells, encompassing both CD4[+] and CD8[+] subsets, was significantly reduced in the cKO mice compared with WT (Fig 6D–I), demonstrating that FAM21 cKO from CD11c[+] dendritic cells of mice impacts multiple innate immune cell functions, and downstream B- and T-cell activation, upon *C. albicans* infection. There is no major difference in the i.p. immune cell population between WT and KO mice before infection with *C. albicans*, supporting that deletion of Fam21 from CD11c-positive cells elicits little or no developmental defects in vivo (Fig S8). Both $T_H$-1 and $T_H$-17 are important immune responses against *C. albicans* (Lin et al, 2009). The FACS data reveal very limited activation of the $T_H$-17 immune response in both animal groups on days 1 and 3 p.i. (Fig S9).

### Transfer of WT BMDCs in vivo partially protects FAM21-cKO mice against *C. albicans* infection

To demonstrate that the enhanced virulence of *C. albicans* in FAM21-cKO mice was indeed due to impaired CD11c[+] dendritic cell function, we performed an adoptive transfer experiment whereby in vitro–cultured WT BMDCs were transferred i.p. into either WT or FAM21-cKO mice 24 h before *C. albicans* infection. After transfer, the WT and cKO mice were infected i.p. with $2.5 \times 10^7$ *C. albicans*, and then, weight loss and survival were monitored. As shown in Fig 7A–D, the FAM21-cKO mice subjected to mock transfer (PBS injection) presented significant weight loss (Fig 7B) and lethality (Fig 7D), whereas the cKO mice pretreated with WT BMDCs were well protected and all survived (Fig 7B and D). Control WT mice that received either PBS or WT-BMDCs presented minimal weight loss (Fig 7A) and all survived after infection (Fig 7C). Accordingly, we conclude that FAM21 is critical to how CD11c[+] dendritic cells function to protect against *C. albicans* infection.

## Discussion

FAM21 is expressed in many mouse organs (Fig S1C), and our endeavors to generate constitutive KO mice were unsuccessful because of embryonic lethality (Fig S1B). We observed that FAM21 is also expressed in BMDCs, which mediate important innate immune functions such as pathogen sensing, phagocytosis, and cytokine secretion, and antigen presentation to activate adaptive immune responses (Flannagan et al, 2012; Eisenbarth, 2019; Cabeza-Cabrerizo et al, 2021). Therefore, we generated FAM21-cKO mice with *FAM21* gene deletion in the CD11C[+] dendritic cell population. Previous studies have linked several components of the WASH complex to human diseases (Zavodszky et al, 2014; Phillips-Krawczak et al, 2015; Turk et al, 2017; Song et al, 2018), so our

---

*C. albicans* and zymosan particles. **(F, G)** ELISA of IL-1$\beta$ (F) and TNF-$\alpha$ (G) levels in the supernatant of BMDCs at 16 h after being treated with HKMT, *C. albicans*, LPS, and Pam3SK4 (n = 3 per genotype). **(H)** Graphical representation of signaling pathways affected in FAM21 cKO BMDCs compared with WT. The scale bar represents 5 $\mu$m. Data are represented as mean ± SD. ns, not significant; *$P$ < 0.05.
Source data are available for this figure.

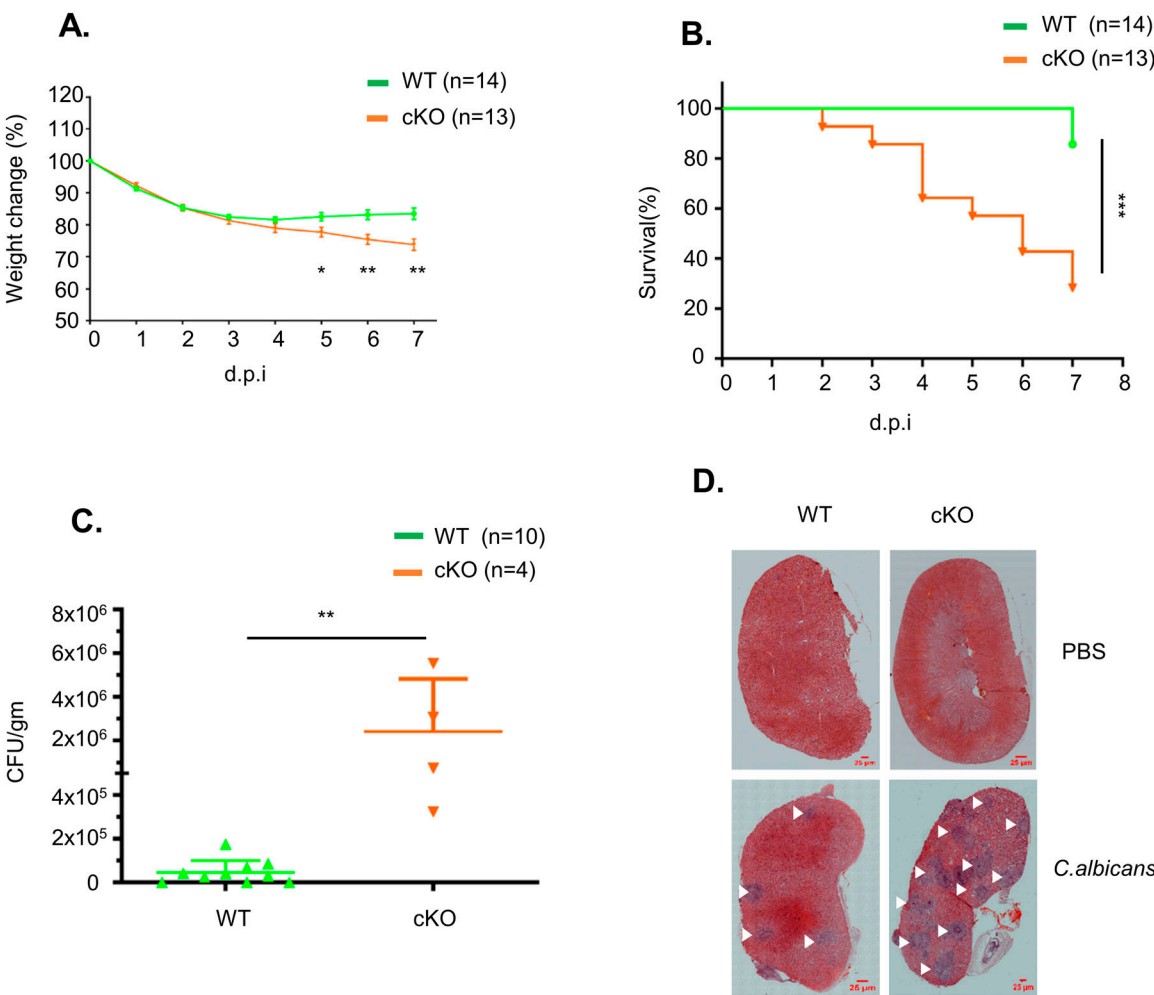

**Figure 5. FAM21-cKO mice are susceptible to *C. albicans*.**
**(A)** Percentage of body weight change in WT and FAM21-cKO mice after i.p. infection with 2.5 × 10⁷ *C. albicans* (n = 14 for WT, n = 13 for cKO). Data are represented as the mean ± SEM. **(A, B)** Survival curves of the same mice described in (A). **(C)** Titers of *C. albicans* in the kidneys of infected mice at 7 d.p.i. (n = 10 for WT, n = 4 for cKO). Data are represented as the mean ± SD. **(D)** Representative H&E staining of kidneys removed from infected WT and cKO mice at 7 d.p.i. The scale bar represents 25 μm. White arrows mark the lesions caused by *C. albicans*. *P < 0.05; **P < 0.01; and ***P < 0.001.
Source data are available for this figure.

FAM21-cKO mice may serve as a useful animal model for further exploring human diseases and the future development of respective therapeutics. There is some ambiguity over the definition of the bone marrow–derived population generated in the presence of GM-CSF as mentioned in Helft et al (2015), as this study considers GM-CSF–derived cells a heterogenous mixture of cells and needs further sorting of cells before using, whereas other researchers (Lutz et al, 2016) consider there is no need for modification; however, few recent studies (Belabed et al, 2020; Matsumura et al, 2021) continued to use the previous approach of using GM-CSF–derived BMDC to study the function of BMDCs. Overall, we agree this aspect of the study can be carefully taken into consideration in future studies.

Our in vitro characterization of FAM21-cKO BMDCs revealed their altered cell morphology and migratory capability. Those cells also displayed enlarged endosomes, consistent with previous data on FAM21-knockdown HeLa cells (Hsiao et al, 2015). Most importantly,

our FAM21-cKO BMDCs exhibited a reduced ability for pathogen internalization and protein antigen presentation, even though external peptide antigen presentation remained intact. Interestingly, our FAM21-cKO BMDCs also exhibited a reduced ability to activate the TLR2/CLEC4E signaling pathway in vitro, and FAM21-cKO mice challenged with *C. albicans* lost more body weight and displayed greater mortality when compared to WT mice. Our analyses of immune cell recruitment in vivo during the course of *C. albicans* infection of FAM21-cKO mice revealed fewer neutrophils and macrophages at the infection site at day 1 p.i., and fewer B and T cells at day 3 p.i. Reconstituting FAM21-cKO mice with WT CD11c⁺ BMDCs rescued the cKO mice from death caused by *C. albicans* infection, supporting that FAM21 is important for TLR2/CLEC4E-mediated signaling in CD11c⁺ dendritic cells for host protection against *C. albicans*.

We found that some phenotypic differences of FAM21-cKO cells relative to WT could be linked to WASH complex functions. For

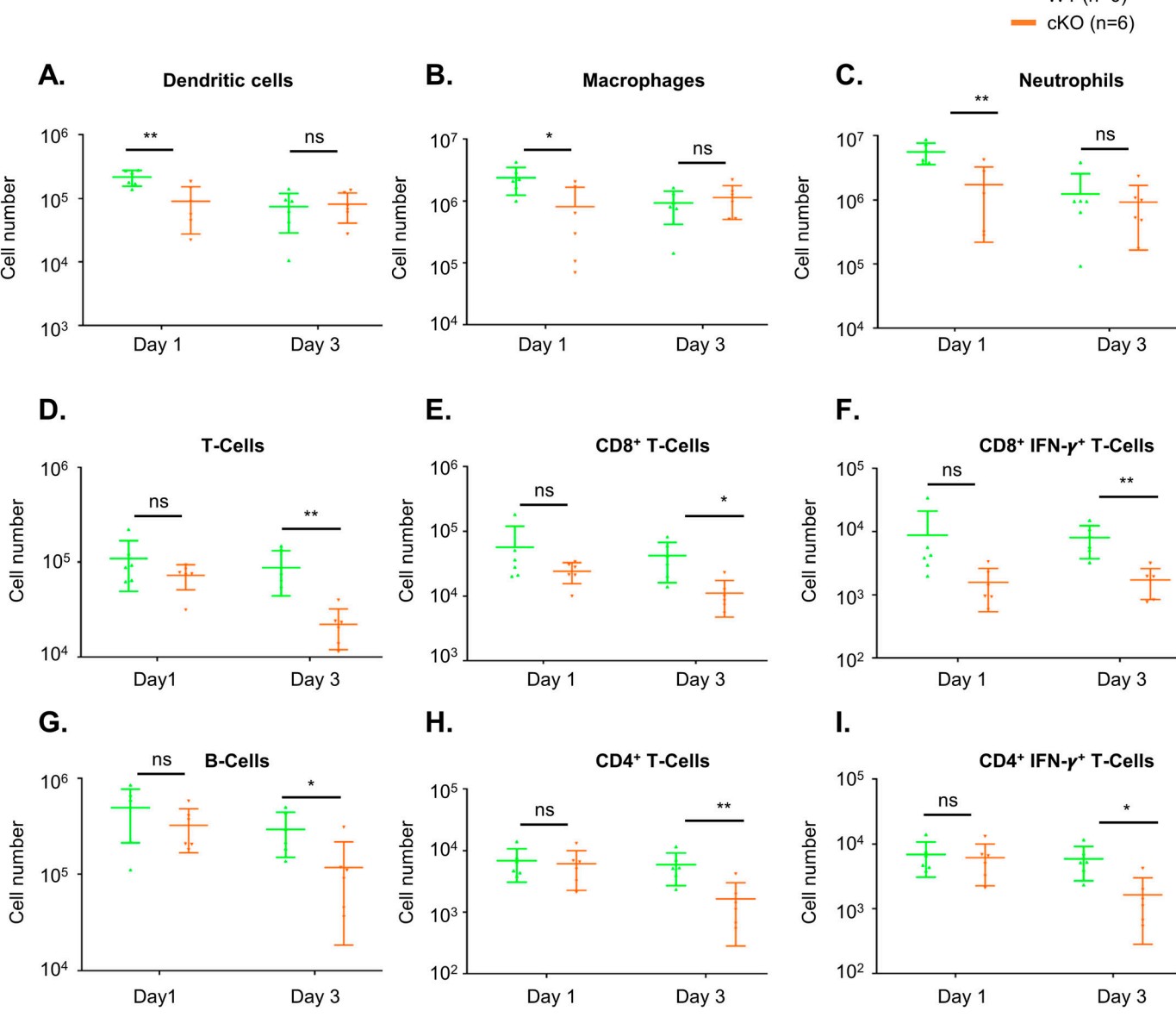

**Figure 6. FAM21-cKO mice display defective innate and adaptive immune responses at the i.p. site of *C. albicans* infection.**
Flow cytometry of i.p. immune cell composition isolated from WT and cKO mice, as described in Fig S7, at 1 and 3 d.p.i. with *C. albicans*. **(A, B, C, D, E, F, G, H, I)** Dendritic cells, (B) macrophages, (C) neutrophils, (D) total T cells, (E) CD8⁺ T cells, (F) CD8⁺IFN-γ⁺ T cells, (G) B cells, (H) CD4⁺ T cells, and (I) CD4⁺IFN-γ⁺ T cells (n = 6 mice per genotype). Data are represented as the mean ± SEM. ns, not significant; *P < 0.05; and **P < 0.01.
Source data are available for this figure.

example, WASH and ARP2/3 regulate early endosome fission, and WASH KO cells were found to possess enlarged early endosomes (Duleh & Welch, 2010), that is, similar to our finding for FAM21-cKO cells. Moreover, integrins are known to mediate cell adhesion and phagocytic functions (Lukacsi et al, 2017; Torres-Gomez et al, 2020), and an accumulation of integrins in the macrophage lysosomes of *Drosophila* WASH mutants resulted in cell spreading defects (Nagel et al, 2017), again similar to our data herein on FAM21-cKO BMDCs. We noticed that levels of WASH protein, but not retromer subunits VPS35 and VPS26, were somewhat reduced in the FAM21-cKO BMDCs (Fig 1C), consistent with the above-described characterizations.

We also report some unique phenotypes of FAM21-cKO cells that differ from previous studies. Graham et al (2014) showed that WASH KO from dendritic cells resulted in reduced surface levels of MHC class II and that the WASH-KO dendritic cells exhibited a diminished peptide antigen presentation capability (Graham et al, 2014). In contrast, our FAM21-cKO BMDCs displayed enhanced surface expression of MHC class II antigen and peptide antigen presentation ability, relative to WT BMDCs. Thus, FAM21 and WASH may regulate MHC class II recycling independently of each other or they function at different stages of the recycling pathway. Details of the underlying mechanism remain to be established.

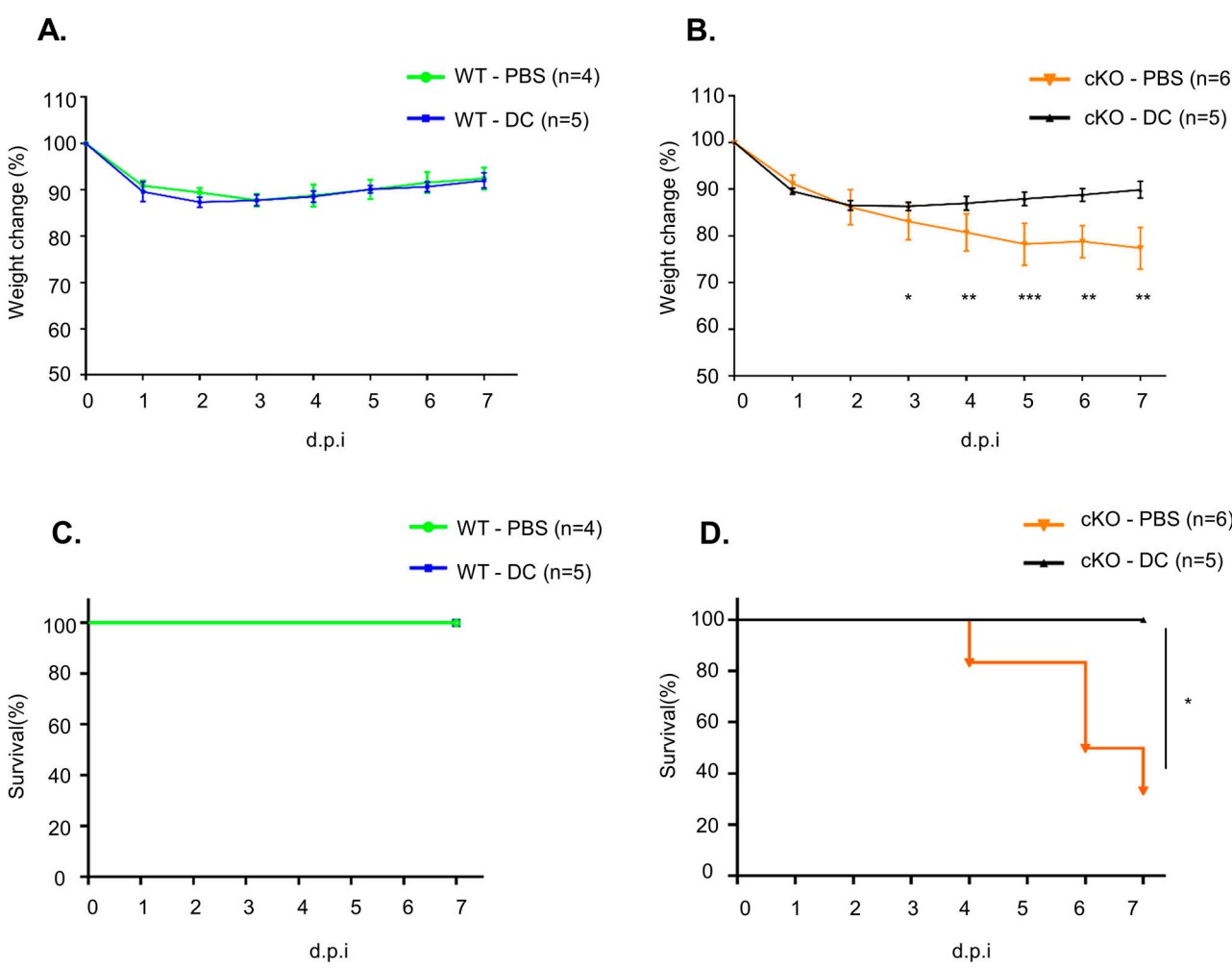

**Figure 7. WT BMDC transfer rescues FAM21-cKO mice from the consequences of *C. albicans* infection.**
**(A, B)** Weight change in (A) WT and (B) FAM21-cKO mice that were i.p. administered with PBS or 5 × 10⁵ WT BMDCs (DC) before *C. albicans* infection. Data are represented as the mean ± SD. **(A, B, C, D)** Survival curves at 7 d.p.i. of (C) WT mice described in (A, D) cKO mice described in (B) (n represents numbers of mice in each group). *$P < 0.05$; **$P < 0.01$; and ***$P < 0.001$.
Source data are available for this figure.

A previous study reported that FAM21 in pancreatic cells may regulate NF-κB target gene transcription via chromatin binding in cell nuclei (Deng et al, 2015). Interestingly, CLEC4E has been reported as a target gene of the transcription factor C/EBPβ (Matsumoto et al, 1999). Hence, it is tempting to speculate that besides its impact on NF-κB, impairment of the CLEC4E signaling pathway in FAM21-cKO BMDCs may be attributable to reduced chromatin binding of C/EBPβ to the promoter region of the *CLEC4E* gene. It will be interesting to elucidate whether FAM21 regulates transcription factor interaction with chromatin in a WASH-independent manner. Overall, our study highlights the importance in vivo of FAM21 in host protection against pathogen invasion.

## Materials and Methods

Our initial attempt to generate a constitutive FAM21 KO mouse line failed because the respective KO mouse embryos exhibited abnormal development at embryonic day (E) 7.5, implying that FAM21 is essential for embryogenesis (Fig S1A and B). Accordingly, we altered strategy and generated FAM21-cKO mice using a BAC cloning system (RP23-1904), obtained with the assistance of the Transgenic Core Facility of the Institute of Molecular Biology, Academia Sinica. We used the Red/ET BAC recombination system from Gene Bridges (Cat. nos. K004 [loxP] and K005 [FRT]). Two loxP sites were inserted into introns 1 and 6 of FAM21, and a Frt-Neo-Frt selection cassette was then inserted after the loxP site in intron 1 according to the manufacturer's instructions. We depict the targeting vector construct in Fig S1D, in which 10 kbps of the *FAM21* gene sequence (accession number: ENSMUSG00000024104) spanning exons 2–6 and encoding three in-frame methionine start codons were removed via Cre recombination. The resulting BAC construct was electroporated into C57BL/6 embryonic stem (ES) cells (Open Biosystems). Three independent ES cell clones carrying the targeted allele were microinjected into albino C57BL/6J (#00058; Jackson Laboratory) blastocysts to obtain FAM21^f-Neo/+ germline-

transmitted mice, which were then crossed with ACTB-FLPe mice to remove the Neo cassette. The resulting $FAM21^{f/+}$ mice were then either mated with EIIa-Cre$^+$ mice to obtain $FAM21^{+/-}$ or were intercrossed with $FAM21^{f/+}$ mice to obtain $FAM21^{f/f}$. These latter homozygous $FAM21^{f/f}$ floxed mice were crossed with CD11c-Cre$^+$ mice (kindly provided by Dr. Nan-Shih Liao, Institute of Molecular Biology, Academia Sinica, Taipei, Taiwan [#008068; JAX stock]) (Caton et al, 2007) to generate the $FAM21^{f/+}$;CD11c-Cre$^{Tg/+}$ mice, which were then crossed with $FAM21^{f/f}$ mice to obtain the $FAM21^{f/f}$; CD11c-Cre$^{Tg/+}$ line. Subsequently, $FAM21^{f/f}$;CD11c-Cre$^{Tg/+}$ mice were mated with $FAM21^{+/-}$ to generate offspring of four different genotypes, that is, $FAM21^{f/+}$, $FAM21^{f/+}$;CD11c-Cre$^{Tg/+}$, $FAM21^{f/-}$, and $FAM21^{f/-}$;CD11c-Cre$^{Tg/+}$ (Fig S1E).

## Animal welfare

All animal protocols were approved by the Institutional Animal Care and Use Committee of Academia Sinica and were conducted in strict accordance with the guidelines on animal use and care of the Taiwan National Research Council's Guide. Animals were euthanized with carbon dioxide at the end-point of the experiment, and all precautions were taken to minimize animal suffering throughout the study.

## BMDC cell culture

Murine BMDCs were cultured according to established protocols as described previously (Lutz et al, 1999), but with minor modifications. In brief, bone marrow cells were isolated from the femur and tibia of 6- to 8-wk-old WT and KO mice, counted, and seeded for 7 d at $2 \times 10^6$ cells/10 cm dish in bacterial culture plates in 10 ml conditioned medium (RPMI supplemented with 10% FBS [HyClone], 1% penicillin–streptomycin [Gibco], 2 mM L-Glutamine, 1 mM sodium pyruvate, 1% non-essential aas [Gibco], 20 mM Hepes, 50 $\mu$m 2-mercaptoethanol, and 10% growth supernatant of GM-CSF–transduced J558L cells) (Qin et al, 1997). We replenished 10 ml of conditioned medium on day 3, and 50% of the medium was replaced with fresh medium on day 6. Non-adherent cells from the culture were collected on day 8 to analyze BMDC culture purity and for in vitro experiments. Generally, the cultured BMDCs comprised 80–85% CD11b$^+$ CD11c$^+$ cells, of which 10–15% were a CD86$^+$ MHC class II$^+$ cell population according to flow cytometric analyses.

## Reagents and antibodies

Details of all reagents used in this study are presented in Table 1.

## Flow cytometry of BMDCs

Flow cytometry of BMDCs was performed as described in Yin et al (2015), but with some modifications. In brief, $5 \times 10^5$ BMDCs were preincubated for 20 min on ice with supernatant collected from hybridoma clone 2.4G2 cell culture (Unkeless, 1979) to block non-specific binding of Fc $\gamma$ receptor, followed by staining for 20 min on ice with a fluorescent antibody cocktail containing anti-CD11c-PE (BioLegend), anti-CD11b-APC-eFluor 780 (eBioscience), anti-CD86-APC (eBioscience), and anti-MHC class II–PB (BioLegend). After

washing three times with FACS buffer (PBS containing 2% FBS) and resuspension in FACS buffer, the BMDCs were analyzed by flow cytometry (BD LSR-II; BD Biosciences).

## Immunoblot analyses

Immunoblot analyses were performed as described previously (Kasani et al, 2017). In brief, $2 \times 10^6$ BMDCs were collected, washed three times with PBS, and lysed in 100 $\mu$l sample buffer (4% SDS, 20% glycerol, 125 mM Tris, pH 6.8, 0.25% bromophenol blue, and 200 mM $\beta$-mercaptoethanol). Cell lysates were heated and separated by SDS–PAGE, transferred to nitrocellulose membrane, and probed with antibodies recognizing FAM21 (1:1,000; Merck), WASH (1:1,000; Abcam), VPS29 (1:1,000; IMGENEX), VPS35 (1:1,000; Sigma-Aldrich), $\beta$-actin (1: 2,000; BioVision), and CD11c-biotin (1:1,000). To detect spleen tyrosine kinase (SYK) phosphorylation, $2 \times 10^6$ BMDCs were infected with *C. albicans* at a ratio of 1:10 (cells:*C. albicans*) for 15 min at 37°C and then washed three times with PBS. Cells were lysed using protein lysis buffer before loading equal amounts of protein (15 $\mu$g) per lane for SDS–PAGE. After transfer to nitrocellulose membrane, the membranes were blocked in 5% BSA and incubated overnight at 4°C with primary anti-phospho-SYK (1:1,000; Cell Signaling Technologies) or anti-SYK (1:1,000; Cell Signaling Technologies) antibodies. The blots were then washed three times with PBST (PBS containing 0.1% Tween-20), incubated at room temperature for 1 h with HRP goat anti-rabbit IgG (1:20,000; Jackson), and then developed using a Western Lightning Enhanced Chemiluminescence kit (PerkinElmer) according to the manufacturer's protocol.

## Immunofluorescence imaging analyses

Immunofluorescence staining was performed as described previously (Izmailyan et al, 2012). In brief, $2 \times 10^5$ BMDCs were seeded overnight on a glass coverslip in a 12-well plate with conditioned medium. The next day, the cells were fixed with 4% PFA at room temperature for 15 min, blocked with 1% BSA (Sigma-Aldrich) in PBS, and then stained with anti-EEA1 (1:250; Santa Cruz), anti-FAM21 (1:1,000; Merck), anti-phalloidin–TRITC (1:1,000; Sigma-Aldrich), and anti-$\alpha$-tubulin (1:1,000; Amersham) antibodies for 1 h at room temperature in PBS containing 0.25% saponin and 1% BSA. The BMDCs were then stained with Cy5-donkey anti-goat IgG (Jackson, 1:1,000), Cy3-donkey anti-rabbit IgG (H+L) (1:2,000; Jackson), and FITC-goat anti-mouse IgG-Fab-specific (1:1,000; Sigma-Aldrich) secondary antibodies for 1 h at room temperature. Nuclei were stained with DAPI for 5 min. Images were taken using a Zeiss LSM 780 confocal microscope with a 63× objective lens. Early endosomes were quantified using the software Imaris 9.8 (Oxford Instruments). The particle area ($\mu$m$^2$) of each endosome was measured, and the size of all early endosomes for both WT and cKO BMDC (~4,000 endosomes per genotype) was drawn as a violin plot.

## Migration assay

The BMDC cell migration assay was performed as described previously (Scandella et al, 2004; Toki et al, 2013). In brief, $1 \times 10^6$ BMDCs in 100 $\mu$l medium (RPMI + 1% FBS) were seeded in a well at the upper chamber of a 12-well transwell plate (3241; Costar), and 600 $\mu$l medium (RPMI + 1% FBS) containing the chemokine CCL19

**Table 1.  Reagents and antibodies.**

| Reagent name | Company | Catalog number |
|---|---|---|
| Anti-WASH complex subunit FAM21C antibody | Merck | ABT79 |
| Anti-VPS26 antibody | Abcam | ab23892 |
| Anti-VPS35 antibody | IMGENEX | IMG-3575 |
| Anti-WASH1 antibody | Sigma-Aldrich | SAB4200372 |
| Anti-$\beta$-actin antibody | BioVision | 3598-100 |
| Anti-EEA1 antibody | Santa Cruz | SC-6415 |
| Anti-$\alpha$-tubulin antibody | Amersham Life Science | N356 |
| Phalloidin–tetramethylrhodamine B isothiocyanate | Sigma-Aldrich | P1951 |
| Anti-phospho-SYK (Tyr525/526) antibody | Cell Signaling Technology | 2711 |
| Anti-SYK antibody | Cell Signaling Technology | 2712 |
| Anti-CLEC4E/Mincle antibody | MBL | D292-3 |
| Anti-paxillin antibody | BD Transduction Laboratories | 610051 |
| Cy5-donkey anti-goat IgG (H+L) | Jackson Laboratories | 705-175-147 |
| Cy3-donkey anti-rabbit IgG (H+L) | Jackson Laboratories | 711-166-152 |
| FITC-goat anti-mouse IgG-Fab-specific | Sigma-Aldrich | F8771 |
| Recombinant murine MIP-3$\beta$ (CCL19) | PeproTech | 250-27B |
| pHrodo Red *E.coli* BioParticles | Molecular Probes | P35361 |
| Zymosan A *S.cerevisiae* BioParticles, Texas Red conjugate | Molecular Probes | Z2843 |
| Il-$\beta$ mouse–uncoated ELISA kit | Invitrogen | 88-7013 |
| Mouse TNF-$\alpha$ DuoSet ELISA | R&D Systems | DY410-05 |
| Fixable Viability Dye eFluor 506 | eBioscience | 65-0866 |
| Heat-killed *Mycobacterium tuberculosis* | InvivoGen | tlrl-hkmt-1 |
| Lipopolysaccharides from *Escherichia coli* 0111:B4 | Sigma-Aldrich | L4391-1MG |
| Pam3CSK4 | InvivoGen | tlrl-pms |
| DAPI, FluoroPure grade | Thermo Fisher Scientific | D21490 |
| Albumin from chicken egg white | Sigma-Aldrich | A5503-10 G |
| Dextran–fluorescein | Invitrogen | D1820 |
| PE-anti-mouse CD11c antibody | BioLegend | 117308 |
| Pacific Blue anti-mouse I-A/I-E antibody | BioLegend | 107620 |
| Anti-mouse CD11b APC-eFluor 780 antibody | eBioscience | 47-0112-82 |
| CD86 (B7-2) monoclonal antibody (GL1), APC | eBioscience | 17-0862-82 |
| PE/Cyanine7 anti-mouse TCR $\beta$ chain antibody | BioLegend | 109222 |
| PerCP/Cyanine5.5 anti-mouse CD19 antibody | BioLegend | 115534 |
| Pacific Blue anti-mouse CD8a antibody | BioLegend | 100725 |
| PE/Cyanine7 anti-mouse CD4 antibody | BioLegend | 100422 |
| APC anti-mouse IFN-$\gamma$ antibody | BioLegend | 505810 |
| PerCP/Cyanine5.5 anti-mouse NK-1.1 antibody | BioLegend | 108728 |
| PerCP/Cyanine5.5 anti-mouse CD3$\varepsilon$ antibody | BioLegend | 100328 |
| APC anti-mouse F4/80 antibody | BioLegend | 123116 |
| FITC anti-mouse Ly-6C antibody | BioLegend | 128006 |
| PE/Cyanine7 anti-mouse Ly-6G antibody | BioLegend | 127618 |
| Alexa Fluor 488 anti-mouse IL-17A antibody | BioLegend | 506910 |
| ROR gamma (t) monoclonal antibody (B2D), PE | eBioscience | 12-6918-82 |
| CD45.2 monoclonal antibody (104), PE-eFluor 610 | eBioscience | 61-0454-82 |

(PeproTech) (300 ng/ml) was added to the lower chambers. The plates were incubated at 37°C for 3 h, and the number of cells that migrated through the membrane into the lower compartment was counted.

## Phagocytosis assay

Phagocytosis assay was performed by placing $2.5 × 10^5$ BMDCs in 100 μl on ice for 30 min and then transferring them to an Eppendorf tube. The medium was removed by centrifugation, and then, the cells were incubated for 0–90 min at 37°C with 100 μl of pHrodo Red *E.coli* (P35361; Thermo Fisher Scientific). The BMDCs were washed three times with PBS and resuspended in FACS buffer, before being subjected to flow cytometry (BD LSR-II; BD Biosciences).

## Dextran uptake assay

WT and cKO BMDCs were incubated with dextran–fluorescein (10,000 MW; Invitrogen) at a concentration of 1 μg/ml for 30 and 60 min at 37°C, washed three times with PBS, and then fixed with 4% PFA. Next, the cells were stained with anti-EEA1 and DAPI, before being analyzed as described above in immunofluorescence imaging analyses. Uptake of dextran–fluorescein was quantified using ImageJ software. The total intensity of the dextran–fluorescein taken up by cells was measured and was divided by the number of cells in each photograph to obtain average dextran–fluorescein taken up per cell (~40 cells were analyzed per time point for each genotype).

## Antigen presentation assay

Antigen presentation assay was performed as described previously (Wang et al, 2010) with some modifications. In brief, $1 × 10^5$ BMDCs were pulsed with ovalbumin protein (25 μg/ml) or with ovalbumin-derived OVA 257–264 peptide (SIINFEKL, 10 pm) for 16 h, washed three times with PBS, and then co-cultured with CD8[+] T cells that were purified from spleen and lymph nodes of the OT-1 transgenic mouse and labeled with CFSE at a concentration of 0.5 μM for 10 min (Pulle et al, 2006). Pulsed BMDC and CFSE-labeled CD8[+] T cells co-cultured at a ratio of 1:5 for 48 h were subsequently washed with FACS buffer, stained with anti-CD8-APC antibodies, and analyzed by flow cytometry (BD LSR-II; BD Biosciences) to reveal CFSE dye dilution on CD8[+] T cells.

## RNA extraction and labeling for microarray analysis

RNA was extracted from WT and cKO BMDC cultures using the TRIzol (Invitrogen) method according to the manufacturer's protocol. RNA quality assessment and labeling was performed as described previously (Kasani et al, 2017). In brief, 10 μg of RNA was reverse-transcribed with aminoallyl-modified dUTP using a Superscript Plus Indirect cDNA labeling system (Invitrogen). The cDNA was then coupled with Alexa Fluor dye after purification using a Qiagen column. WT cDNA was labeled with Alexa Fluor 555, and cKO cDNAs were labeled with Alexa Fluor 647. These labeled cDNAs were hybridized to an Agilent SurePrint G3 Mouse GE 8 × 60 K microarray (G4852A) according to the manufacturer's protocol. The microarrays were scanned on an Agilent DNA microarray scanner (Cat. no.:

US9230696) using the two-color scan setting for 8 × 60 K array slides. The scanned images were analyzed in Feature Extraction Software 10.5.1.1 (Agilent) using default parameters (protocol GE2_105_Dec08 and Grid 074809_D_F_20150624) to obtain background- and dye-Norm (linear Lowess) signal intensities.

## Microarray and bioinformatic analyses

The raw microarray data (three biological replicates) were imported into GeneSpring software (version 12.6.1; Agilent). A total of 62,976 probes were then mapped to corresponding genes using the NCBI mm9 Mouse Genome Assembly (NCBI37; Jul2007). After removing probes associated with unnamed genes, non-coding genes, or pre-dicted genes, we were left with a total of 41,442 probes associated with 23,787 genes. Next, the probes with a signal intensity >70 in at least two out of three replicates were selected to test whether the $\log_2$ ratio of cKO/WT signals was significantly different from 0. Under the criteria of a *t* test *P*-value < 0.05 and a $\log_2$ fold change >1.5, we were left with 261 significant differentially expressed probes representing 246 genes. Of these, 104 probes representing 98 genes were up-regulated, whereas 157 probes representing 148 genes were down-regulated in the FAM21-cKO mice. Next, we conducted gene ontology and KEGG pathway enrichment analysis on each set of genes using *clusterProfiler* (v 3.18.1) in the R package (Yu et al, 2012). Finally, the 148 down-regulated genes were analyzed further for predicted and known protein–protein interactions using STRING (version 11.0) (Szklarczyk et al, 2018).

## qRT–PCR analysis

RNA was isolated from WT and cKO BMDCs by the TRIzol (Invitrogen) method. cDNAs were prepared from 1 μg RNA according to the manufacturer's protocol. Quantitative real-time polymerase chain reaction (qRT–PCR) was performed on cDNAs using iQ SYBR Green Supermix (Bio-Rad) in a CFX-96 Touch system (Bio-Rad). Primer sequences are presented in Table 2.

## Cytokine ELISA

WT and cKO BMDCs were seeded in a 96-well U-bottom plate ($5 × 10^5$ cells/well) and stimulated for 16 h at 37°C with each of the following ligands: LPS (10 ng/ml), heat-killed *Mycobacterium tuberculosis* (5 μg/ml), PamCysSerLys (Pam3CSK4, 0.5 μg/ml), or *C. albicans* at a ratio 1:10 (cells:*C. albicans*) in 200 μl conditioned growth medium. Supernatants were collected and subjected to IL-1β (Invitrogen) and TNF-α (R&D Systems) ELISA to detect those cytokines according to the manufacturer's protocols.

## *C. albicans* uptake assay

We cultured $2 × 10^5$ BMDCs overnight on a glass slide in a 12-well plate and then incubated them for 15 or 30 min at 37°C with CFSE-labeled *C. albicans* (ATCC 90028) (Dagher et al, 2018) at a ratio of 1:10 (cells:*C. albicans*). Glass slides were then washed three times with PBS and fixed with 4% PFA. The cells were stained for phalloidin–TRITC (1:1,000) and DAPI, and images were acquired using a Zeiss LSM 780 confocal microscope with a 63× objective lens.

**Table 2.  DNA primer sequences used in the study.**

| Primer name | Primer sequence |
| --- | --- |
| TLR2 F | 5'-CTTCCTGAATTTGTCCAGTACAGGG-3' |
| TLR2 R | 5'-TCGACCTCGTCAACAGGAGAAGGG-3' |
| TNF-$\alpha$ F | 5'-TGTCTCAGCCTCTTCTCATT-3' |
| TNF-$\alpha$ R | 5'-GCCATTTGGGAACTTCTCAT-3' |
| CLEC4E F | 5'-ACCAAATCGCCTGCATCC-3' |
| CLEC4E R | 5'-CACTTGGGGGTTTTTGAAGCATC-3' |
| IL-1$\beta$ F | 5'-CAAATCTCGCAGCAGCACAT-3' |
| IL-1$\beta$ R | 5'-CCACGGGAAAGACACAGGTA-3' |
| CXCL2 F | 5'-ATGGCCCCTCCCACCTGC-3' |
| CXCL2 R | 5'-TCAGTTAGCCTTGCCTTTGTT-3' |
| GAPDH F | 5'-GCTCACTGGCATGGCCTTC-3' |
| GAPDH R | 5'-CCTGCTTCACCACCTTCTTGA-3' |
| MMP13 F | 5'-GTGACTTCTACCCATTTG-3' |
| MMP13 R | 5'-GCAGCAACAATAAACAAG-3' |
| ADAMDEC-1 F | 5'-GTAATTGAGGCTAAAAAAAAGAAT-3' |
| ADAMDEC-1 R | 5'-GCGTGGCCCAACTCATG-3' |

### In vivo infection with *C. albicans* and i.p. immune cell analysis

Male and female 8- to 9-wk-old mice were i.p infected with $2.5 \times 10^7$ *C. albicans* per mouse in 500 $\mu$l PBS as described previously (Bergeron et al, 2017; Ashizawa et al, 2019) and then monitored for disease progression by determining body weight loss. At 1, 3, and 6 d.p.i, mice were euthanized and their i.p. wash was harvested for immune cell population analysis by flow cytometry as described previously (Ray & Dittel, 2010). Mice were euthanized when their body weight had been reduced by >25% of their original body weight on the day of infection. The i.p. wash cell suspension was treated with ammonium–chloride–potassium (ACK) lysis buffer to lyse the erythrocytes, followed by staining with a cocktail of fluorescent antibodies for myeloid and lymphoid cell populations (Table 1). Cells were analyzed by flow cytometry (BD LSR-II; BD Biosciences).

### Histopathology

Mouse kidney histology was assessed as described previously (Kulkarni et al, 2021) with some modifications. The kidneys of WT and cKO mice were removed on 7 d.p.i and fixed in 4% PFA for 2 d at 4°C. The kidneys were then embedded, sectioned, and stained with hematoxylin and eosin (H&E), followed by microscopic examination. Images were photographed using a Zeiss Axio Imager Z1 microscope with 20× objective lenses.

### BMDC transfer for in vivo complementation

We harvested $5 \times 10^5$ WT BMDCs from culture on day 8 and transferred them i.p. in 100 $\mu$l PBS into WT and cKO mice 24 h before *C. albicans* infection, as described previously (Burgess et al, 2014). The next day, mice were infected i.p. with $2.5 \times 10^7$ CFU *C. albicans* per mouse. Mouse body weight was monitored for 7 d to observe the effect of cell complementation on *C. albicans* infection.

### Internalization assay

Internalization assay was performed as described in Sullivan & Coscoy (2008) with modifications; briefly, WT and cKO BMDC cells were stained with fluorescent conjugated anti-CD11c and anti-CD11b antibodies for 30 min on ice and washed with ice-cold FACS buffer to remove unbound antibody. The temperature was raised to 37°C to allow internalization of the antibody, and at each time point, cells were removed and suspended in neutral (pH 7.2) or acidic (pH 1.5) FACS buffer and quenched immediately in neutral buffer. Samples were stored on ice and analyzed by flow cytometry for the internalization of antibodies. Percentage of antibody internalization = $100 \times (S_{tx}-S_{t0})/(T_{t0}-S_{t0})$, $S_{tx}$ is MFI of cells with acidic treatment at each time point, $S_{t0}$ is the MFI of cells with acidic treatment at time 0, and $T_{t0}$ is the MFI of cells with neutral buffer treatment.

### Adhesion assay protocol

Adhesion assay was performed according to Lazarovici et al (2020) with modifications; briefly, $4 \times 10^5$ WT and cKO BMDC cells in conditioned medium were seeded in a fibronectin-coated (10 $\mu$g/ml) or uncoated 96-well plate (SpectraPlate-96HB) at different time points and were washed with the serum-free medium before fixing with 4% PFA. The plates were then stained with crystal violet solution for 30 min and washed with PBS. Finally, the crystal violet was dissolved with 2% SDS solution and O.D readings were taken at 570 nm.

### Statistical analyses

Statistical analyses were performed using a *t* test in GraphPad Prism software (version 9.0), and mouse survival curves were generated using a Mantel–Cox test. *P*-values <0.05 were considered statistically significant: *$P < 0.05$; **$P < 0.01$; and ***$P < 0.001$.

# Data Availability

The data supporting the findings of this study are available within the article and its supplementary materials. Microarray analysis data have been deposited to NCBI GEO with the GEO accession number: GSE196465.

# Supplementary Information

# Acknowledgements

We thank Sue-Ping Lee of the Imaging Core Facility and Ya-Min Lin of the FACS Core Facility for the technical support at the Institute of Molecular Biology, Academia Sinica, Taipei, Taiwan. The work is supported by grants provided by Academia Sinica and Ministry of Science and Technology (110-2320-B-001-015-MY3).

## Author Contributions

R Kulkarni: conceptualization, data curation, formal analysis, validation, investigation, visualization, methodology, and writing—original draft, review, and editing.

SK Kasani: conceptualization, data curation, validation, investigation, visualization, methodology, and writing—original draft.

C-Y Tsai: resources, data curation, validation, investigation, visualization, methodology, and writing—original draft.

S-Y Tung: data curation, software, formal analysis, supervision, investigation, methodology, and writing—original draft.

K-H Yeh: data curation, software, formal analysis, validation, and methodology.

C-HA Yu: software, formal analysis, validation, methodology, and writing—original draft.

W Chang: conceptualization, resources, supervision, funding acquisition, validation, methodology, project administration, and writing—original draft, review, and editing.

## Conflict of Interest Statement

The authors declare that they have no conflict of interest.

## Results

FAM21 is important for CD11c[+] dendritic cell morphology and migration. Moreover, FAM21 is critical for phagocytosis and antigen presentation function of these cells. Finally, FAM21 is required for the activation of the TLR2/CLEC4E pathway and host protection against *C.albicans* infection.

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
