## [Reviewer comments · Life Science Alliance]

Life Science Alliance

FAM21 is critical for TLR2-CLEC4E-mediated dendritic cell function against *Candida albicans*

Rakesh Kulkarni, Siti Kasani, Ching-Yen Tsai, Shu-Yun Tung, Kun-Hai Yeh, Chen-Hsin Yu, and Wen Chang
DOI: <https://doi.org/10.26508/lsa.202201414>

Corresponding author(s): Wen Chang, Institute of Molecular Biology, Academia Sinica

Review Timeline:

Submission Date:	2022-02-17
Editorial Decision:	2022-03-23
Revision Received:	2022-12-01
Editorial Decision:	2023-01-10
Revision Received:	2023-01-17
Accepted:	2023-01-18

Transaction Report:

March 23, 2022

Re: Life Science Alliance manuscript #LSA-2022-01414-T

Dr. Wen Chang
Institute of Molecular Biology, Academia Sinica
5F
128, Sec. 2, Academia Rd., Nankang
Taipei 11529
Taiwan

Dear Dr. Chang,

Thank you for submitting your manuscript entitled "FAM21 is critical for TLR2-CLEC4E-mediated dendritic cell function against *Candida albicans*" to Life Science Alliance. The manuscript was assessed by expert reviewers, whose comments are appended to this letter. We invite you to submit a revised manuscript addressing the Reviewer comments.

Thank you for this interesting contribution to Life Science Alliance. We are looking forward to receiving your revised manuscript.

Sincerely,

B. MANUSCRIPT ORGANIZATION AND FORMATTING:

Reviewer #1 (Comments to the Authors (Required)):

The manuscript by Kulkarni and Khadijah Kasani explored the role of FAM21 a component of the Wiskott Aldrich Syndrome protein and scar homologue complex in dendritic cells. The WASH complex has been identified as a type I nucleation promoting factor, and has been shown to promote actin polymerisation.

The authors showed that germline deletion of FAM21 is embryonically lethal (in line with its broad expression in tissue) and therefore used a conditional ablation system to study its role in dendritic cells. Using mainly GM-CSF produced "DCs?", the authors argued that FAM21 deletion resulted in impaired adhesion, migration. FAM21 deletion also resulted in the formation of larger early endosomes, although this observation remained mainly unexplored. A limited number of functional assays revealed that antigen processing by GM derived DCs is impaired while direct presentation is not. Microarray analysis highlighted a few differentially expressed genes in GM derived DC lacking FAM21 and pointed to a putative role for FAM21 in regulating TLR2/Clec4e signalling pathway.

Major points

- 1- Given the heterogeneity of the culture system used by authors (Inaba et al JEM 1992, Helft et al. Immunity 2015), the nature of the cells claimed to be DC is questionable.
- 2- A detailed analysis on sorted monocyte derived DCs vs mac vs granulocytes will be required in order to assess the role of FAM21 in moDCs and moMacs. Most of the assays presented in this manuscript are blurred by the heterogeneous nature of the cells produced in-vitro. It does appear FAM21 ablation affects moDCs differentiation (MHCII+ CD86+), thus justifying the need to work on purified population.
- 3-Careful analysis of the cellular content of the peritoneal cavity (PC) is lacking (CD11c is not only expressed in DCs) in particular following immune challenge (PC mac, neutrophils...). The manuscript missed the opportunity to address the role of FAM21 in the development of DCs in steady state? This is unfortunate given the authors have the tools at hand and hinders their capacity in drawing conclusion from their in vivo challenge with *Candida albicans*.
- 4-The justification for using germline deletion of FAM21 is lacking. Adding to that, the introduction in its current form lacks some rationale on why studying the role of FAM21 in DCs might be important? The reader is left with some rather wide statement (lane 102).

Additional points:

- cell associated antigen presentation should be monitored (OVA coated splenocytes)
- lack of acknowledgements/references regarding where the Cre deleter are coming from?
- the method to quantify the size of the endosome is lacking?
- Standard adhesion assay should be provided
- Migration to CCL19 appears to be high, in particular in absence of stimulation (e.g. TLR agonism)

Reviewer #2 (Comments to the Authors (Required)):

Kulkarni and collaborator examine the role of FAM21 in antigen presentation by dendritic cells. FAM21 is part of the WASH complex that regulates endocytosis and intracellular trafficking. They show that FAM21 cKO BMDCs display a defect in antigen internalization and cross-presentation in vitro. Furthermore, they find a reduced expression of CLEC4E at the RNA level and

surface expression level that is associated with a defect in *C. Albicans* uptake and sensing by FAM21 cKO BMDCs. Next the authors show that FAM21 cKO mice are more susceptible to *C. Albicans* infection. This was associated with reduced immune cell recruitment to the injection site. Transfer WT BMDCs in FAM21 cKO mice rescued this phenotype, suggesting an important role of DCs in controlling infection. The authors convincingly show a role for FAM21 in antigen uptake by dendritic cells and although they do not formally show it their data suggest that FAM21 cKO DCs have a different phenotype from WASH KO DCs. However, I find that the authors could be a little more careful regarding their conclusions that are not always fully supported by their data. The manuscript could also benefit from some rewriting.

Figure EV1E: Could we author clarify why "that strategy generated KO mice unsuited to in vivo experiments". Line 118

Fig1A: It seems that there is a higher proportion of CD86^{med}/MHCII^{med} cells in the FAM21 KO BMDCs in comparison to the in the WT BMDCs. than in the WT. Could the authors quantify the three populations CD86^{low}/MHCII^{low}, CD86^{med}/MHCII^{med}, CD86^{high}/MHCII^{high}.

Fig1C: Have multiple splicing variants of *Fam21* been previously described or is it purely speculative?

"This outcome reflects a fission defect of endosomes in cKO cells, similar to our previous observation of FAM21 cKO from HeLa cells" The authors should rephrase this sentence since they did not show per se that the FAM21 cKO BMDCs phenotype actually results from a fission defect.

Fig1I: Could the authors quantify the CD11c band.

The authors observe a reduced surface expression of CD11b and CD11c and speculate that it is related to a recycling defect in FAM21 BMDCs, but wouldn't they expect an increase in CD11b and CD11c surface expression? They authors could easily measure CD11c and CD11b internalization by FACS.

Fig2C: Can the authors quantify dextran uptake?

Fig2D: Is the increased in MHCII and CD86 in FAM21 BMDCs related to reduced internalization of the molecules?

Fig2E: Is the increased OT-1 stimulation by FAM21 BMDCs the result of an increased surface MHCI in comparison to WT BMDCs?

Fig2F: The authors conclude "Thus, cKO BMDCs displayed a reduced antigen processing ability, even though peptide antigen presentation on cell surfaces remained intact". The authors have no data to suggest a defective antigen processing. The reduced OT-1 stimulation by FAM21 BMDCs is likely the result of a reduced uptake as shown in fig 2A and 2C.

Fig. 4 Th1 and Th17 immune immunity is critical for protective immunity against *C. Albicans*. Is MHCII presentation and CD4 T cell stimulation altered in FAM21 BMDCs? This could be easily be addressed using BMDCs in vitro.

Line 147: replace "Fibronectin" by "Fibronectin"

Reviewer #1 (Comments to the Authors (Required)):

The manuscript by Kulkarni and Khadijah Kasani explored the role of FAM21 a component of the Wiskott Aldrich Syndrome protein and scar homologue complex in dendritic cells. The WASH complex has been identified as a type I nucleation promoting factor, and has been shown to promote actin polymerisation.

The authors showed that germline deletion of FAM21 is embryonically lethal (in line with its broad expression in tissue) and therefore used a conditional ablation system to study its role in dendritic cells. Using mainly GM-CSF produced "DCs?", the authors argued that FAM21 deletion resulted in impaired adhesion, migration. FAM21 deletion also resulted in the formation of larger early endosomes, although this observation remained mainly unexplored. A limited number of functional assays revealed that antigen processing by GM derived DCs is impaired while direct presentation is not. Microarray analysis highlighted a few differentially expressed genes in GM derived DC lacking FAM21 and pointed to a putative role for FAM21 in regulating TLR2/Clec4e signalling pathway.

Major points

1- Given the heterogeneity of the culture system used by authors (Inaba et al JEM 1992, Helft et al. Immunity 2015), the nature of the cells claimed to be DC is questionable.

2- A detailed analysis on sorted monocyte derived DCs vs mac vs granulocytes will be require in order to assess the role of FAM21 in moDCs and moMacs. Most of the assays presented in this manuscript are blurred by the heterogeneous nature of the cells produced in-vitro. It does appear FAM21 ablation affect moDCs differentiation (MHCII+ CD86+), thus justifying the need to work on purified population.

Response for 1 & 2:

The reviewer raised the study by Helft *et al.* 2015, in which the authors stated that GM-CSF-derived dendritic cultures (BMDC) comprise moDC and moMacs. Nevertheless, other studies (e.g., Lutz *et al.*, 2016) have argued against the need for further purification. Furthermore, recent studies (Belabed *et al.*, 2020; Matsumura *et al.*, 2021) have continued to use the same method to study BMDC biology that we deployed. As ever more surface markers are used to study the BMDC population, it is conceivable that small subsets of cell types will be identified. While we understand the reviewer's concern, we remain confident that our data solidly support all our conclusions.

3-Careful analysis of the cellular content of the peritoneal cavity (PC) is lacking (CD11c is not only expressed in DCs) in particular following immune challenge (PC mac, neutrophils....). The manuscript missed the opportunity to address the role of FAM21 in the development of DCs in

steady state? This is unfortunate given the authors have the tools at hand and hinders their capacity in drawing conclusion from their in vivo challenge with Candida albicans.

Response: We have now included the data requested by the reviewer (Figure EV7). The new data shows that there is no major difference in the intraperitoneal immune cell population between wild-type and KO mice before infection with *C. albicans*, supporting that deletion of Fam21 from CD11c-positive cells elicits little or no developmental defects *in vivo* (Lines: 267-269, page:12).

4-The justification for using germline deletion of FAM21 is lacking. Adding to that, the introduction in its current form lacks some rationale on why studying the role of FAM21 in DCs might be important? The reader is left with some rather wide statement (lane 102).

Response: Our aim was to study the role of Fam21 *in vivo*, so initially we tried more than one approach to obtain *FAM21*-KO mice, either using whole-body or tissue-specific knockout strategies. Upon uncovering that Fam21 is essential for mouse embryonic development, we logically modified our research direction to investigate the functions of Fam21 in innate immune cells such as BMDCs since innate immune cell development is dispensable for embryonic development. Thus, we could obtain viable offspring to study the *in vivo* function of Fam21 in mice.

In the revised manuscript, we now explain why we chose to knock out the *FAM21* gene in innate immune dendritic cells (DC) (in lines: 114-121, page: 6).

Additional points:

- cell associated antigen presentation should be monitored (OVA coated splenocytes)

Response: We performed OVA antigen uptake assays that clearly demonstrated an important role for Fam21 in antigen uptake and presentation *in vitro*. Therefore, we do not think this experiment is necessary to support our conclusion.

- lack of acknowledgements/references regarding where the Cre deleter are coming from?

Response: Apologies for this oversight. The CD11c-Cre mice were kindly provided by Dr. Nan-Shih Liao, Institute of Molecular Biology, Academia Sinica, Taipei, Taiwan (JAX stock #008068) (Caton *et al.*, 2007) (lines: 369-371, page:16).

- the method to quantify the size of the endosome is lacking?

Response: Apologies, the quantification method is now described in the revised manuscript (lines: 440-442, page: 19).

- Standard adhesion assay should be provided

Response: We have performed the standard adhesion assay as requested by the reviewer and the resulting data show an adhesion defect for cKO cells (Figure EV6).

- Migration to CCL19 appears to be high, in particular in absence of stimulation (e.g. TLR agonism)

Response: We followed the methodology described in the study by (Toki *et al.*, 2013) and our results are consistent with their data. Based on those results, we concluded that the cKO cells migrated faster than the wild-type cells.

Reviewer #2 (Comments to the Authors (Required)):

Kulkarni and collaborator examine the role of FAM21 in antigen presentation by dendritic cells. FAM21 is part of the WASH complex that regulates endocytosis and intracellular trafficking. They show that FAM21 cKO BMDCs display a defect in antigen internalization and cross-presentation in vitro. Furthermore, they find a reduced expression of CLEC4E at the RNA level and surface expression level that is associated with a defect in C. Albicans uptake and sensing by FAM21 cKO BMDCs. Next the authors show that FAM21 cKO mice are more susceptible to C. Albicans infection. This was associated with reduced immune cell recruitment to the injection site. Transfer WT BMDCs in FAM21 cKO mice rescued this phenotype, suggesting an important role of DCs in controlling infection. The authors convincingly show a role for FAM21 in antigen uptake by dendritic cells and although they do not formally show it their data suggest that FAM21 cKO DCs have a different phenotype from WASH KO DCs. However, I find that the authors could be a little more careful regarding their conclusions that are not always fully supported by their data. The manuscript could also benefit from some rewriting.

Figure EV1E: Could we author clarify why "that strategy generated KO mice unsuited to in vivo experiments". Line 118

Response: Reviewer 1 has also posed a similar question. Germline deletion of *FAM21* generated viable mice containing one functional allele of the *FAM21* gene in every cell in the body, so any KO phenotype in the germline deletion mice could not be attributed to a defect in a specific cell type.

To make our strategy clearer, we have added the following text to the revised manuscript (lines: 122-131, page: 6):

We generated *FAM21*^{+/-} germline deletion mice in which one allele of the *FAM21* gene had been knocked out in all somatic cells. Crossing the *FAM21*^{+/-} mice with *FAM21*^{ff} *CD11c-Cre* mice generated cKO BMDCs hosting only residual Fam21 protein, representing a suitable null mutant for *in vitro* studies. However, these germline *FAM21*^{+/-} mice are unsuited to study Fam21 function *in vivo* because they generate offspring containing somatic cells with only one functional allele of the *FAM21* gene. Therefore, for our purposes, we bred *Fam21*^{ff} mice with *CD11c-Cre* mice to specifically knock out the *FAM21* gene from *CD11c*-positive dendritic cells, allowing us to address the biological roles of Fam21 *in vivo* (Figure EV1F). All the mice we generated according to these two KO strategies developed normally (data not shown).

In the revised manuscript, we also explain why we chose to knock out the *FAM21* gene in innate immune dendritic cells (DC) (lines: 114-121, page:6):

“DC play an essential role in innate immunity such as pathogen sensing, cytokine secretion, and antigen presentation to activate adaptive immune responses (Cabeza-Cabrerizo *et al.*, 2021; Eisenbarth, 2019). Furthermore, depletion of immune cells or KO of genes in the immune system often does not cause lethality in mice, allowing us to obtain adult KO mice for phenotypic studies. We also rationalized that many DC functions rely on cytoskeleton rearrangement and cell polarization, which may be involved in the cargo trafficking activity modulated by Fam21.”

Fig1A: It seems that there is a higher proportion of CD86^{med}/MHCII^{med} cells in the FAM21 KO BMDCs in comparison to the in the WT BMDCs. than in the WT. Could the authors quantify the three populations CD86^{low}/MHCII^{low}, CD86^{med}/MHCII^{med}, CD86^{high}/MHCII^{high}.

Response: Yes, we have quantified the CD86^{high}MHCII^{high}, CD86^{med}MHCII^{med} and CD86^{low}MHCII^{low} populations (Figure EV8) and the data do not show a higher proportion of CD86^{med}/MHCII^{med} cells in the *FAM21*-KO BMDCs relative to wild-type cells.

Fig1C: Have multiple splicing variants of Fam21 been previously described or is it purely speculative?

Response: Multiple splicing variants of *FAM21* have been reported previously (http://asia.ensembl.org/Mus_musculus/Transcript/Summary?db=core;g=ENSMUSG00000024104;r=6:116184999-116239647;t=ENSMUST00000036759;tl=uDStZbFgkwFh7i2y-7980460), as shown in the schematic of Figure EV2. However, whether these splicing variants explain the varying sizes of *FAM21* protein detected in our immunoblot (Figure 1C) is speculative.

"This outcome reflects a fission defect of endosomes in cKO cells, similar to our previous observation of FAM21 cKO from HeLa cells" The authors should rephrase this sentence since they did not show per se that the FAM21 cKO BMDCs phenotype actually results from a fission defect.

Response: The sentence has been modified to:

“We speculate that this outcome could be derived from an endosome fission defect in the cKO cells, similar to our previous observation for *FAM21*-KO HeLa cells (Hsiao *et al.*, 2015). (lines: 151-153, page: 7)”

Fig1I: Could the authors quantify the CD11c band.

Response: Yes, we have quantified the CD11c bands using Image J, and the ratio of relative intensity between the wild-type and cKO bands is 1:0.93.

The authors observe a reduced surface expression of CD11b and CD11c and speculate that it is related to a recycling defect in FAM21 BMDCs, but wouldn't they expect an increase in CD11b

and CD11c surface expression? They authors could easily measure CD11c and CD11b internalization by FACS.

Response: We have now performed internalization assays for both Cd11b and CD11c (Figure EV5) and observed faster internalization kinetics for the cKO BMDCs when compared to wild-type cells. We have modified the respective text to:

“We performed the internalization assay twice for the CD11c and CD11b receptors, respectively, by means of flow cytometry and observed that both receptors were internalized faster in the cKO BMDCs relative to the WT cells (Figure EV5). Despite this difference in internalization kinetics, we cannot rule out a contribution of other recycling defects to our observations for the *FAM21*-KO cells (lines: 167-172, page:8).”

Fig2C: Can the authors quantify dextran uptake?

Response: We have now quantified dextran uptake and present the data in Figure 2D. Please note that we have also updated our Materials & Methods section accordingly (lines: 462-466, page:20).

Fig2D: Is the increased in MHCII and CD86 in FAM21 BMDCs related to reduced internalization of the molecules?

Response: It is indeed possible that an increase of surface level CD86 and MHC-II could be related to trafficking defects, such as internalization or recycling. We now discuss this possibility in our revised manuscript (lines: 140-141, page:7).

Fig 2E: Is the increased OT-1 stimulation by FAM21 BMDCs the result of an increased surface MHCI in comparison to WT BMDCs?

Response: We did not measure surface expression levels of MHC-I in wild-type or KO BMDC cells, but it has previously been reported that increased expression of MHC-I molecules in DCs results in enhanced peptide antigen presentation (Ackerman & Cresswell, 2003). Accordingly, it is possible that the enhanced ability of our cKO BMDCs to present peptide antigen is due to increased levels of MHC-I molecules on cell surfaces (lines: 200-205, page:9).

Fig2F: The authors conclude "Thus, cKO BMDCs displayed a reduced antigen processing ability, even though peptide antigen presentation on cell surfaces remained intact". The authors have no data to suggest a defective antigen processing. The reduced OT-1 stimulation by FAM21 BMDCs is likely the result of a reduced uptake as shown in fig 2A and 2C.

Response: We agree with the reviewer that the reduced T activation displayed by cKO BMDCs could be due to reduced antigen uptake (Figure 2F). Hence, we have modified the text accordingly (lines: 199-200, page:9).

Fig. 4 Th1 and Th17 immune immunity is critical for protective immunity against C.Albicans. Is MHCII presentation and CD4 T cell stimulation altered in FAM21 BMDCs? This could be easily

be addressed using BMDCs in vitro.

Response: Yes, both T_H-1 and T_H-17 are important immune responses against *C. albicans* (Lin *et al.*, 2009). In fact, in our original manuscript, we already presented *in vivo* data (Figure 6F and I) to show that the T_H-1 immune response was better induced in wild-type mice relative to *FAM21*-KO mice. We have now also added our *in vivo* data for the T_H-17 immune response in mice upon infection with *C. albicans* (Figure EV9). The FACS data reveal very limited activation of the T_H-17 immune response in both animal groups on days 1 and 3 post-infection (lines: 270-272, page:12).

Line 147: replace "Fibronectin" by "Fibronectin"

Response: Apologies, now corrected.

January 10, 2023

RE: Life Science Alliance Manuscript #LSA-2022-01414-TR

Dr. Wen Chang
Institute of Molecular Biology, Academia Sinica
5F
128, Sec. 2, Academia Rd., Nankang
Taipei 11529
Taiwan

Dear Dr. Chang,

Thank you for submitting your revised manuscript entitled "FAM21 is critical for TLR2-CLEC4E-mediated dendritic cell function against *Candida albicans*". We would be happy to publish your paper in Life Science Alliance pending final revisions necessary to meet our formatting guidelines.

- please rename your EV figures as supplementary figures and update the figure callouts in the main manuscript text accordingly
- please add a figure callout for Figure 2G and Figure S1D to your main manuscript text
- please make sure that the panels in your supplementary figures are properly explained in the figure legends
- GEO dataset GSE196465 should be made publicly accessible at this point
- please remove the 'Paper Explained' section

Figure Check:

- please place scale bars on the bottom right corner of images for consistency
- please add scale bars to Figure 5D

A. FINAL FILES:

B. MANUSCRIPT ORGANIZATION AND FORMATTING:

Sincerely,

January 18, 2023

RE: Life Science Alliance Manuscript #LSA-2022-01414-TRR

Dr. Wen Chang
Institute of Molecular Biology, Academia Sinica
5F
128, Sec. 2, Academia Rd., Nangang
Taipei 11529
Taiwan

Dear Dr. Chang,

Thank you for submitting your Research Article entitled "FAM21 is critical for TLR2-CLEC4E-mediated dendritic cell function against *Candida albicans*". It is a pleasure to let you know that your manuscript is now accepted for publication in Life Science Alliance. Congratulations on this interesting work.

DISTRIBUTION OF MATERIALS:

Again, congratulations on a very nice paper. I hope you found the review process to be constructive and are pleased with how the manuscript was handled editorially. We look forward to future exciting submissions from your lab.

Sincerely,
